# ATM inhibition exploits checkpoint defects and ATM-dependent double strand break repair in *TP53*-mutant glioblastoma

Daniel J. Laverty[1], Shiv K. Gupta [2], Gary A. Bradshaw [3], Alexander S. Hunter[3], Brett L. Carlson[2], Nery Matias Calmo [1], Jiajia Chen[2,4], Shulan Tian[2], Jann N. Sarkaria [2] & Zachary D. Nagel [1]✉

Determining the balance between DNA double strand break repair (DSBR) pathways is essential for understanding treatment response in cancer. We report a method for simultaneously measuring non-homologous end joining (NHEJ), homologous recombination (HR), and microhomology-mediated end joining (MMEJ). Using this method, we show that patient-derived glioblastoma (GBM) samples with acquired temozolomide (TMZ) resistance display elevated HR and MMEJ activity, suggesting that these pathways contribute to treatment resistance. We screen clinically relevant small molecules for DSBR inhibition with the aim of identifying improved GBM combination therapy regimens. We identify the ATM kinase inhibitor, AZD1390, as a potent dual HR/MMEJ inhibitor that suppresses radiation-induced phosphorylation of DSBR proteins, blocks DSB end resection, and enhances the cytotoxic effects of TMZ in treatment-naïve and treatment-resistant GBMs with *TP53* mutation. We further show that a combination of G2/M checkpoint deficiency and reliance upon ATM-dependent DSBR renders *TP53* mutant GBMs hypersensitive to TMZ/AZD1390 and radiation/AZD1390 combinations. This report identifies ATM-dependent HR and MMEJ as targetable resistance mechanisms in *TP53*-mutant GBM and establishes an approach for simultaneously measuring multiple DSBR pathways in treatment selection and oncology research.

Many cancer therapies, including radiation and alkylating agents, exert their cytotoxic effects by forming double stranded breaks (DSBs) in DNA. Cancer cells that efficiently repair these therapy-induced DSBs can resist the toxic effects of DNA-damaging therapies, causing treatment failure. This has spurred the development of pharmacological DSB repair (DSBR) inhibitors that enhance the antitumor effects of DNA-damaging therapy and overcome treatment resistance[1]. The goal of adding DSBR inhibitors to DNA-damaging therapies is to sensitize cancer cells to treatment while sparing healthy cells; however, to predict which patients will benefit from DSBR inhibitors, we must understand the molecular mechanisms of DSBR in cancer. DSBR is a

multifaceted process involving multiple pathways, and there is presently a dearth of multiplexed methods for assessing DSBR in cancer cell lines and patient samples.

Glioblastoma (GBM) is an aggressive brain cancer treated almost exclusively with radiation and temozolomide (TMZ), both of which kill GBMs mainly by forming DSBs. Radiation directly breaks DNA, while TMZ does so indirectly by forming $O^6$-methylguanine ($O^6$-MeG). This lesion is normally repaired by methylguanine methyltransferase (MGMT); however, approximately half of GBMs silence *MGMT* through promoter hypermethylation[2]. Consequently, $O^6$-MeG persists until replication, where it mispairs with thymidine[3],

[1]Harvard T.H. Chan School of Public Health, Boston, MA 02115, USA. [2]Mayo Clinic, Rochester, MN 55905, USA. [3]Harvard Medical School, Boston, MA 02115, USA. [4]Shengjing Hospital of China Medical University, Shenyang 110004, China. ✉e-mail: znagel@hsph.harvard.edu

triggering futile cycles of mismatch repair (MMR) that eventually create DSBs when replisomes collide with MMR intermediates[3,4]. Despite the central role of DSBs in GBM therapy, a holistic view of how multiple DSBR pathways relate to treatment response is lacking. As targeted agents continue to fail in the clinic, DSBR inhibitors that sensitize GBMs to frontline therapies are an increasingly attractive option[5–7].

Human cells have two major repair pathways for DSBs: non-homologous end joining (NHEJ) and homologous recombination (HR). NHEJ, the major repair pathway for radiation-induced DSBs, rapidly rejoins breaks with minimal processing[8]. HR is slower than NHEJ but is highly accurate and is especially important for repairing collapsed replication forks[9]. In HR, nucleolytic processing exposes long 3′-single-stranded tails that are paired to homologous sequences on the sister chromatid to prime repair synthesis and break resolution[10]. A third DSBR pathway, microhomology-mediated end joining (MMEJ), sometimes referred to as alternative end joining, involves minimal end resection that exposes short (2–15 bp) complementary sequences (microhomologies) on both sides of the break that are annealed, extended, and ligated[11,12]. DNA polymerase theta (Pol theta, encoded by the *POLQ* gene) is critical for microhomology annealing and fill-in synthesis[12–14], and is essential in HR-defective cancers and in cells with other DSBR defects[5,15,16]. In these cells, Pol theta inhibitors (POLQi) induce cell death and overcome acquired PARP inhibitor resistance driven by upregulated *POLQ*[5,7], highlighting the therapeutic potential of understanding DSBR mechanisms in cancer.

Many GBMs initially respond to treatment; however, resistance is essentially inevitable. Two major TMZ resistance mechanisms have been documented: reactivation of *MGMT* and down-regulation of MMR[17–19], with the latter occurring most frequently in clinical isolates[20,21]. Neither resistance mechanism can currently be targeted therapeutically; however, some TMZ-resistant GBMs lack alterations in MMR or MGMT activity[18], implying that additional resistance mechanisms exist. Prolonged TMZ treatment enhances HR activity in GBM cell lines, and genetic knockdown of HR proteins overcomes acquired TMZ resistance in some GBM cells[22,23]. This suggests that repair of therapy-induced DSBs contributes to GBM treatment resistance and that DSBR inhibitors in conjunction with TMZ may yield new combination therapies. Although clinical-grade HR inhibitors are currently lacking, the recent development of potent inhibitors against other DSBR proteins including Pol theta and the kinase, Ataxia-telangiectasia mutated (ATM), presents attractive opportunities for such combination therapies[6,7]. We investigated DSBR in treatment-resistant GBMs with the goal of identifying new combination therapies employing TMZ with DSBR inhibitors.

We previously reported plasmid-based fluorescence multiplexed host cell reactivation (FM-HCR) assays for multiple DNA repair pathways[24], which advanced our understanding of TMZ response in GBM patient-derived xenografts (PDXs)[18]. Here we report Fluorescence Multiplexed Double Strand Break Repair (FM-DSBR) analysis, a method for simultaneously measuring NHEJ, HR, and MMEJ with plasmid-based reporter assays. Like earlier HCR assays, this approach is amenable to cell lines, primary samples, and xenografts. Using this approach, we identify upregulated HR/MMEJ as a targetable resistance mechanism in GBMs. We screen clinically relevant small molecules and identify the ATM inhibitor (ATMi), AZD1390, as a dual HR/MMEJ inhibitor that enhances the cytotoxicity of TMZ and radiation in GBMs with *TP53* mutation. Our data indicate that *TP53*-mutant GBMs rely upon ATM-dependent HR/MMEJ to repair treatment-induced DSBs, and in the presence of AZD1390, undergo cell death when unrepaired DSBs persist into mitosis.

## Results

### MMEJ efficiency is enhanced in GBM PDX samples with acquired TMZ resistance

We first hypothesized that MMEJ is a targetable treatment-resistance mechanism in GBM, so we generated a plasmid-based MMEJ reporter assay (Fig. 1A). Using a similar approach to previous genomically-integrated assays[11], we inserted a restriction site and flanking 8 bp microhomologies into the BFP reporter gene, which abolished fluorescence in cells transfected with non-linearized plasmid (Fig. S1). When the plasmid was linearized in vitro and transfected into cells, we detected BFP fluorescence that was decreased by *POLQ* knockdown or inhibition in the GBM cell line, U251 (Fig. 1B), by *POLQ* knockdown in U2OS (Fig. S1), or by *POLQ* knockout in TK6 (Fig. S2). Confident in our ability to detect MMEJ in live cells, we investigated the role of this pathway in treatment response in GBM.

We first treated U251 cells with TMZ, which enhanced BFP_MMEJ8 activity in a dose-dependent manner (Fig. 1C), suggesting that GBM cells use MMEJ to resist the cytotoxic effects of TMZ. Consistent with this interpretation, *POLQ* depletion sensitized U251 to killing by TMZ in clonogenic survival assays (21% survival of si*POLQ* cells at 10 μM TMZ vs. 43% in siNT, Fig. 1D, E). To interrogate whether MMEJ activity is associated with TMZ resistance in patient-derived samples, we measured MMEJ in four GBM PDX cultures (G12, G14, G22, and G39) and their counterparts with acquired TMZ resistance (denoted "-TMZ")[25]. We simultaneously measured MMR and MGMT activity using FM-HCR, as we hypothesized that MMEJ repairs TMZ-induced DSBs that occur in MGMT-null, MMR-proficient GBMs.

MMEJ efficiency was significantly increased in three of four TMZ-resistant clones (TMZ2296, TMZ5476, and TMZ8023) derived from G12 (Fig. 1F). Interestingly, MGMT activity remained absent in these three resistant clones, and MMR was only modestly reduced compared to parental G12 (Fig. 1G, H). Conversely, clone G12-TMZ3080, which functionally restored MGMT (Fig. 1H, third bar from left), showed no change in MMEJ (Fig. 1F, third bar from left). These data suggest that MMEJ mainly contributes to TMZ resistance under conditions of replication fork collapse (MGMT-null, MMR-proficient cells). Further supporting this model, we observed increased MMEJ in only one other resistant line, G22-TMZ, where MGMT activity remained absent and MMR activity was only modestly reduced compared to parental G22. Conversely, the resistant line G39-TMZ, which displayed a marked reduction in MMR (Fig. 1G), showed no significant change in MMEJ (Fig. 1F).

Finally, we investigated repair in the G14/G14-TMZ pair. G14 is MGMT-proficient and is intrinsically resistant to TMZ, so this PDX pair serves as a useful test for the effect of prolonged in vivo TMZ treatment under conditions where $O^6$-MeG is efficiently reversed. Interestingly, MMEJ efficiency did not increase in G14-TMZ and instead was decreased (Fig. 1F). G14 parental line displayed robust MGMT activity (Fig. 1H), which was further increased in G14-TMZ, as was MMR (Fig. 1G). Taken together, these results suggest that MMEJ contributes to TMZ resistance mainly in MGMT-null, MMR-proficient GBMs where $O^6$-MeG is expected to cause fork collapse.

### The POLQi, ART558, does not sensitize GBMs to TMZ

Enhanced MMEJ in TMZ resistant GBMs suggested a targetable resistance mechanism, so we tested TMZ in combination with the POLQi, ART558. Interestingly, ART558 did not enhance cell killing by TMZ in either G22 or G22-TMZ in a viability assay (Fig. 1I). Similar results were obtained in GBM cell lines when assessing viability (A172, U87, SNB75, Fig. S3) or clonogenic survival (U251, Fig. S3). HCR assays revealed that, while ART558 inhibited MMEJ, it caused a concurrent increase in HR (Fig. 1J), which was already elevated at baseline in G22-TMZ (Fig. 1J, 0.71% in G22 vs. 1.5% in G22-TMZ). HR is the major repair pathway for TMZ-induced DSBs[26], and MMEJ makes a comparatively minor

contribution to DSBR in HR-proficient cells[11]. We hypothesized that intact HR limits the efficacy of POLQi in combination with TMZ that inhibiting both HR and MMEJ is required to robustly sensitize GBM cells to TMZ.

We first interrogated this hypothesis using clonogenic survival assays in U251. To benchmark our results against previous reports, we depleted MSH2, which ablates DSB formation by TMZ and causes TMZ resistance[4]. In our hands, siRNA knockdown of MSH2 imparted complete TMZ resistance (Fig. 1K), confirming the fidelity of our approach. We then depleted the HR/MMEJ nuclease, CtIP (encoded by the *RBBP8* gene), which markedly enhanced TMZ toxicity (10% survival of si*RBBP8* cells at 10 μM TMZ vs. 37% in siNT). Knockdown of the NHEJ gene, *LIG4*, had no major effect on TMZ sensitivity (46% survival at 10 μM vs. 37% in siNT, p = 0.16), consistent with previous

results that NHEJ is not involved in TMZ resistance[26]. These data indicate that inhibiting the initiation of HR and MMEJ has therapeutic potential in GBM. To further explore this possibility, we developed a platform for simultaneously measuring multiple DSBR pathways.

## Development of fluorescence multiplexed double strand break repair (FM-DSBR) assay

FM-DSBR comprises a reported NHEJ assay (Fig. 2A)[24] and new assays for HR and MMEJ. Using a similar strategy to ref. 27, we created an HR assay (Cherry_HR) where recombination between genes on two plasmids that do not express fluorescent protein results in mCherry expression (Fig. 2B). We detected Rad51-dependent mCherry signal only when linearized plasmid and homology donor plasmid were co-transfected (Fig. S4), consistent with measurement of HR. We

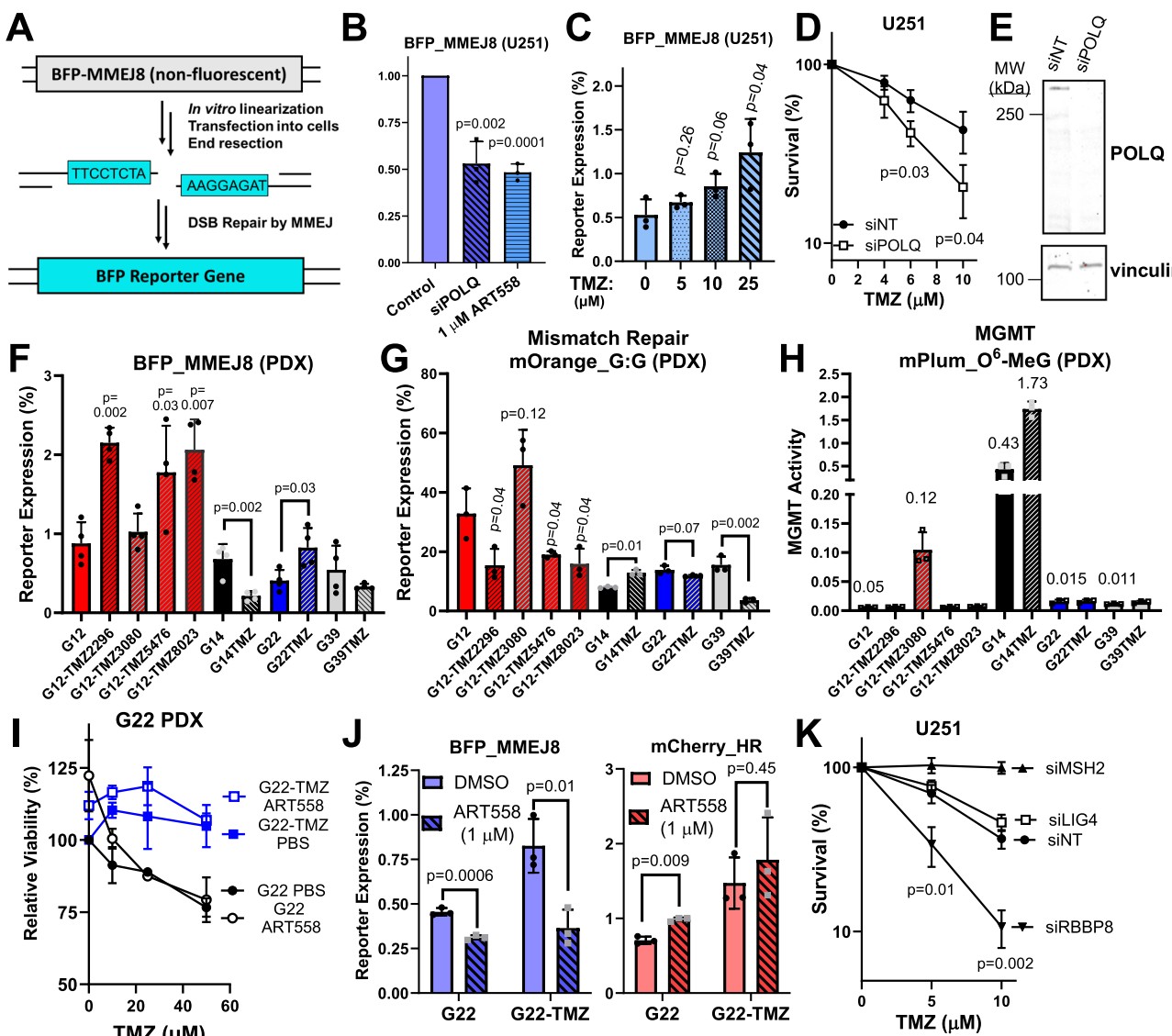

**Fig. 1 | MMEJ activity is increased in GBM models with acquired TMZ-resistance. A** Illustration of BFP_MMEJ8 reporter and (**B**) validation in U251 cells treated with ART558 or *POLQ*-targeting sRNA (si*POLQ*). Reporter expression was calculated as described in methods and normalized to DMSO control for ART558 or non-targeting siRNA (siNT) control for si*POLQ*) (**C**) BFP_MMEJ8 activity in U251 treated with TMZ (48 h), washed, and transfected with BFP_MMEJ8 24 h later. **D** Clonogenic survival of U251 transfected with siNT or si*POLQ*). **E** Western blot from (**D**). **F**−**H** Measurement of MMEJ, MMR, and MGMT activity in PDX lines. "TMZ" denotes acquired TMZ resistant subline (striped bars) derived from the parental line with

the same number. **I** Relative viability (CellTiter Glo 2.0) of G22 and G22-TMZ treated with TMZ for 120 h with or without ART558 (1 μM). **J** MMEJ and HR assays in G22 and G22-TMZ treated as indicated. **K** Clonogenic survival of U251 transfected with indicated siRNA and treated with TMZ (96 h) followed by growth for 7–10 days. Data are presented as the mean of three independent experiments (except for (**F**) where n = 4), error bars show the standard deviation (SD), and p-values are from statistical comparison by unpaired two-tailed t-test. In (**B**−**D**) and (**K**) comparison is to control (DMSO or siNT) and in (**F**, **G**), comparison is to parental PDX line. Source data are provided as a source data file.

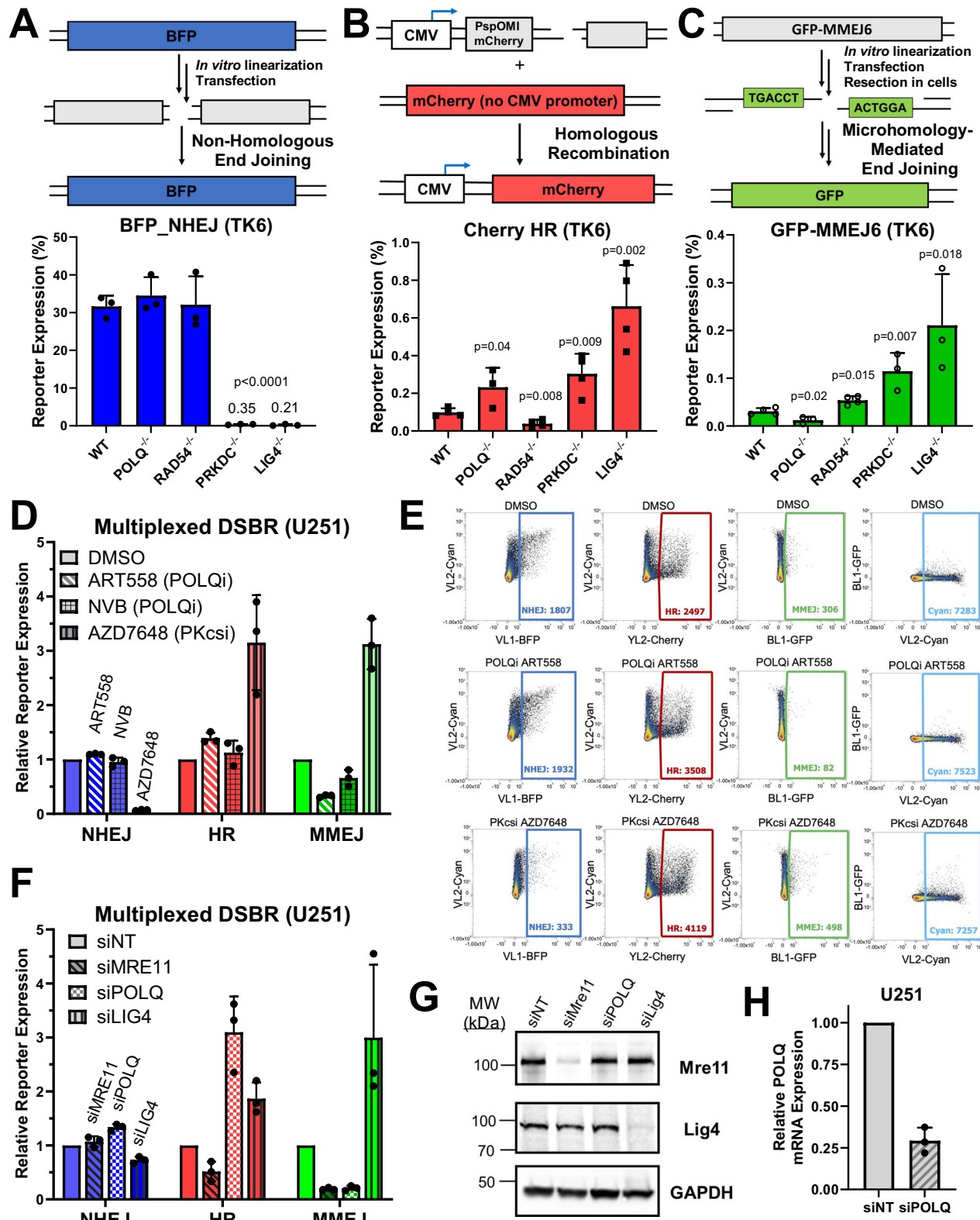

**Fig. 2 | Fluorescence-based multiplexed double strand break repair assays (FM-DSBR) simultaneously measure the activity of three DSBR pathways.** Illustrations of fluorescent reporter plasmids for NHEJ (**A**), HR (**B**), and MMEJ (**C**) and validation in TK6 wild type (WT) and knockout cell lines[78]. **D** FM-DSBR analysis in U251 treated with ART558 (1 μM), NVB (100 μM), or AZD7648 (1 μM) for 4 h prior to transfection. **E** Representative flow cytometry plots from (**D**) indicating reporter expression for the three reporters plus an AmCyan-expressing transfection control plasmid. **F** FM-DSBR in U251 72 h after siRNA knockdowns (siNT: non-targeting siRNA). **G** Western blot of cells in (**F**). The samples derive from the same experiment but different gels for Mre11 and Lig4. These were processed in parallel. **H** Validation of *POLQ* knockdown by qRT-PCR using actin as a control. Data are presented as the mean of n = 3 independent experiments except for (**B**) where n = 4, error bars show SD, and p-values are from comparison to WT by unpaired two-tailed t test. Source data are provided as a source data file.

generated an optimized MMEJ reporter, GFP_MMEJ6 (Figs. 2C and S5), seeking to improve the POLQ-dependency of BFP_MMEJ8, which was relatively weak (Fig. 1B). GFP_MMEJ6 contains 6 bp microhomologies recessed 10 nt from the DSB, based on Pol theta substrates reported by ref. 28, and is more responsive to POLQ knockdown or inhibition than BFP_MMEJ8 (compare Figs. 1B to 2D, F).

We validated each DSBR reporter using cells from the TK6 Consortium (https://www.nihs.go.jp/dgm/tk6.html). Knockout of NHEJ genes *PRKDC* and *LIG4* markedly reduced BFP_NHEJ signal (Fig. 2D). and increased both HR (Figs. 2E, 3-fold for *PRKDC*$^{-/-}$ and 6-fold for *LIG4*$^{-/-}$, respectively) and MMEJ (Fig. 2F, 4-fold and 8-fold increases, respectively). Knockout of the HR gene *RAD54* suppressed HR by 3-fold and enhanced MMEJ by 2-fold. Knockout of *POLQ* reduced GFP_MMEJ6 signal by 2.5-fold and enhanced Cherry_HR by 2-fold. Primary fibroblasts from patients with genetic DNA repair deficiency showed similar results, with decreased NHEJ and increased HR/MMEJ in Lig4 syndrome fibroblasts compared to apparently healthy individuals (Fig. S6). Taken together, these results indicate that DSBR reporters faithfully detect DSBR alterations in cell lines and patient samples.

We combined the three DSBR reporters (BFP_NHEJ, mCherry_HR, and GFP_MMEJ6) with pMax_AmCyan transfection control to yield a fluorescence multiplexed-DSBR analysis method, FM-DSBR. We treated U251 cells with POLQi, ART558 or novobiocin (NVB), or the DNA-PKcs inhibitor, AZD7648. As expected, both POLQi suppressed MMEJ activity (Figs. 2D, E), with ART558 proving more potent than NVB (70% reduction in GFP_MMEJ6 for 1 μM ART558 vs. 40% for 100 μM NVB). ART558 slightly enhanced HR efficiency (30% increase) while NVB showed no appreciable change. The DNA-PKcs inhibitor markedly suppressed NHEJ activity (94% reduction in BFP_NHEJ) and caused a concurrent increase in HR (320% increase) and MMEJ (310% increase), as expected based on previous results[29]. We further validated FM-DSBR using siRNA knockdown (Fig. 2F). Depletion of *MRE11* (Fig. 2G) markedly suppressed both HR (50% decrease) and MMEJ (79% decrease), while depletion of *POLQ* (Fig. 2H) suppressed only MMEJ (85% decrease) and caused a concurrent increase in HR (300% increase). Depletion of *LIG4* attenuated BFP_NHEJ activity (30% decrease) and markedly enhanced HR and MMEJ activity (~200% and 300% increase, respectively). Taken together, these data support the ability of FM-DSBR to simultaneously assess NHEJ, HR, and MMEJ in GBM cells.

## Acquisition of TMZ resistance is associated with defective NHEJ, enhanced HR/MMEJ, and radiosensitivity

Knockdown of *RBBP8* enhanced killing by TMZ in U251, suggesting that resection-dependent DSBR pathways promote TMZ resistance in GBM cell lines. To extend this analysis of DSBR to patient-derived samples, we applied FM-DSBR in G22/G22-TMZ and G59/G59-TMZ pairs and assessed response to TMZ or radiotherapy in orthotopic xenograft models. HR and MMEJ were elevated in G22-TMZ (Fig. 3A), consistent with a role for these pathways in TMZ resistance and reinforcing our findings with BFP_MMEJ8 (Fig. 1F). Interestingly, this was coupled with a significant (p = 0.002) reduction in NHEJ—the main pathway for repair of radiation-induced DSBs[9]—in G22-TMZ (12.5% vs. 8.5%). We found that mice engrafted with G22 did not respond to radiotherapy (median survival 43 days, placebo and RT, Fig. 3B); however, those engrafted with acquired resistant line, G22-TMZ, showed a markedly better response to radiotherapy (median survival 68 days, p < 0.0001, Fig. 3B), suggesting that the NHEJ defect observed by FM-DSBR results in radiosensitivity. Exome sequencing of the G22/G22-TMZ pair revealed a damaging mutation in the *LIG4* gene (Table S1), which results in a P452L amino acid substitution, further supporting our observation that NHEJ is impaired and HR/MMEJ enhanced in G22-TMZ.

We observed a similar phenomenon in an additional PDX pair. NHEJ efficiency was significantly decreased and HR/MMEJ were increased in G59-TMZ compared to G59 parental (Fig. 3D). Mice engrafted with G59 responded to radiotherapy (median survival 74 days) and TMZ (median survival 90 days) compared to placebo (median survival 46 days, Fig. 3E). TMZ had no effect on mice engrafted with G59-TMZ, but radiotherapy markedly improved survival (median survival 343 days, p < 0.0001, Fig. 3F), suggesting that, like G22-TMZ, impaired NHEJ renders G59-TMZ hypersensitive to radiation. Unlike in G22-TMZ, we did not observe mutations in core NHEJ genes, aside from an *XRCC5* mutation resulting in V405I substitution, which is predicted to be tolerated (Table S1). However, we detected an A > T transversion in *RIF1*, resulting in an N2021Y substitution, and two transitions (A > G and T > C) in *PPP1CC*, resulting in F227S and Y225C substitutions, respectively. The products of each gene, Rif1 and protein phosphatase 1, act in the 53BP1 pathway to antagonize DSB end resection, promoting NHEJ and inhibiting HR[30]. Taken together, these PDX data suggest that TMZ monotherapy induces DSBR alterations that impair NHEJ and enhance HR/MMEJ, causing radiosensitivity.

Elevated HR/MMEJ in TMZ-resistant GBM xenografts suggest that these pathways promote TMZ resistance and poor response to TMZ monotherapy. However, most GBM patients receive TMZ/radiation combination therapy instead of TMZ alone, so functional DSBR measurements in PDX samples with acquired TMZ resistance may not reveal clinically relevant resistance mechanisms. We therefore used cBioPortal[31–33] to analyze gene expression in The Cancer Genome Atlas (TCGA) GBM dataset[34] to assess whether signatures of enhanced HR/MMEJ correlate with treatment response. We selected GBM patients receiving TMZ (most of whom also receive radiotherapy), stratified by median expression of DSBR genes, and compared overall survival. Expression of core MMEJ and HR factors including *POLQ* or *BRCA1* was not correlated with GBM patient survival (Fig. 3G and Table S2). Because we identified mutations in members of the 53BP1-Rif-shieldin pathway in the recurrent GBM, G59-TMZ, we hypothesized that expression level of genes in this pathway may be associated with treatment response. Interestingly, we found that expression of *SHLD1* was positively correlated with survival in GBM patients receiving TMZ, while expression of *TRIP13*, which dissociates shieldin to promote end resection[35], was inversely correlated with survival. Similar survival trends were observed for *TRIP13* and *SHLD1* expression in low grade glioma patients receiving TMZ (Fig. S7). These data suggest that gene expression levels consistent with high DSB end resection are associated with poor survival of patients receiving TMZ.

## FM-DSBR identifies clinically relevant dual HR/MMEJ inhibitors

The sensitivity of si*RBBP8* cells to TMZ and the association of elevated HR/MMEJ with TMZ resistance suggest that simultaneous HR/MMEJ inhibition is a promising strategy for GBM combination therapy. We used FM-DSBR to search for a clinical-grade dual HR/MMEJ inhibitor. We tested molecules that cross the blood-brain barrier and were previously reported to inhibit DSBR, including inhibitors of ATM, mTOR, histone deacetylases (HDACs), and bromodomain and extra-terminal domain (BET) family proteins[36–39]. We also included kinase inhibitors such as Sorafenib, Buparlisib, Trametinib, and Ibrutinib to broaden the scope of our screen. These drugs have been tested in GBM but have no known effects on DSBR, so they served as putative negative controls.

We treated U251 cells with each drug for 2 h and transfected FM-DSBR reporters. Importantly, we note that FM-DSBR incorporates a control transfection containing the wild-type (WT) reporter plasmids (pMax_BFP, pMax_GFP, pMax_mCherry, and pMax_AmCyan) to control for drug effects on transfection or reporter gene expression.

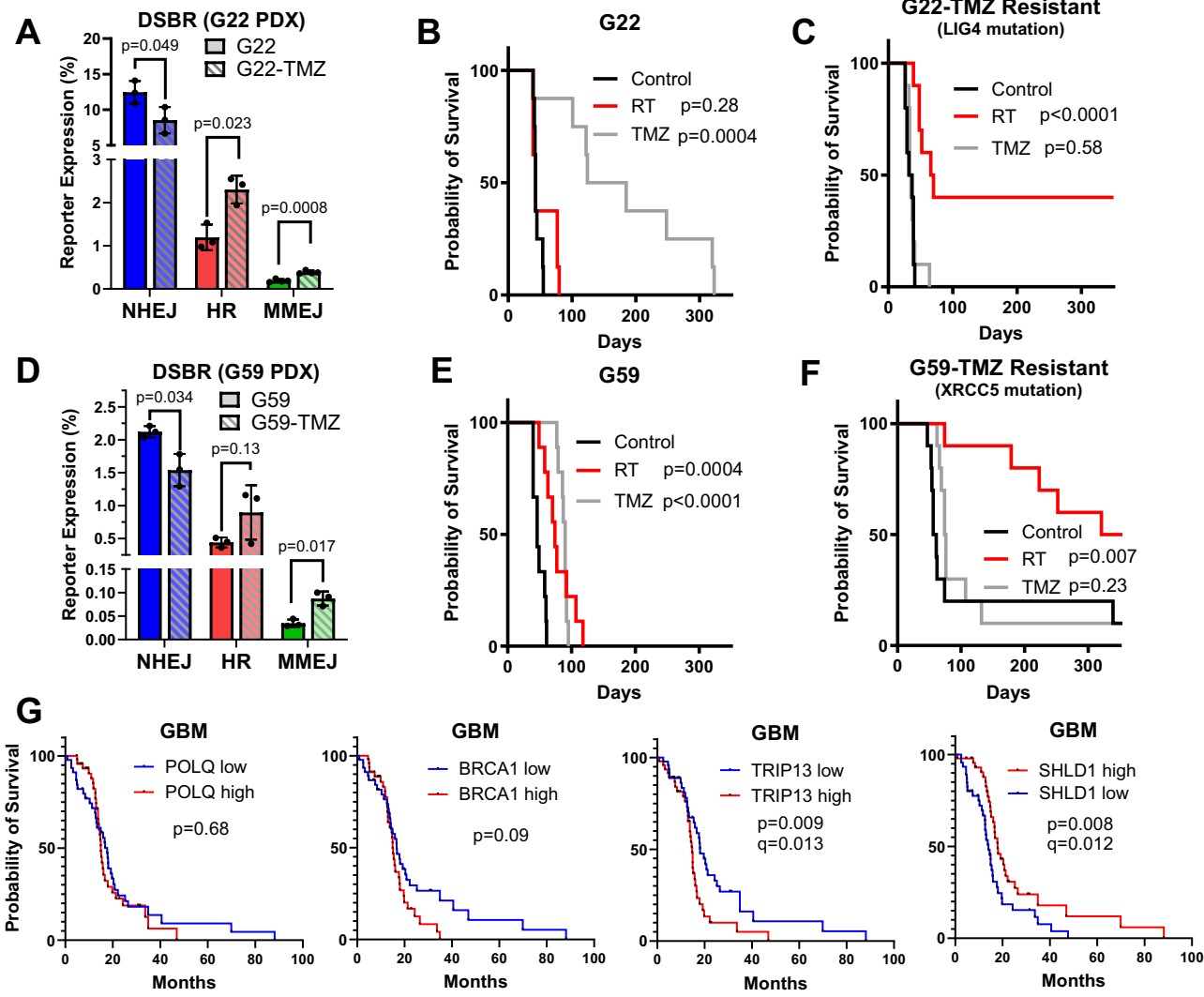

**Fig. 3 | TMZ resistance is associated with altered DSBR, especially enhanced HR and MMEJ. A** FM-DSBR analysis of G22 (parental) and G22-TMZ (acquired TMZ resistance) PDX lines. **B**, **C** Survival analysis of mice ($n = 8$ for G22 and $n = 10$ for G22-TMZ) implanted intracranially with the indicated PDX line and treated with placebo, TMZ, or radiotherapy. **D** FM-DSBR analysis of G59 and G59-TMZ acquired resistant line. **E**, **F** Survival analysis of mice implanted with G59 ($n = 9$) or G59-TMZ ($n = 10$) and treated as described for (**B**, **C**). **G** Survival of GBM patients treated with temozolomide and stratified by median expression of the indicated gene using cBioportal ($n = 46$ per group)[31–33]. In (**A**, **D**), data are presented as the mean of three independent experiments, error bars show SD, and p-values are from multiple unpaired two-tailed t-tests with Holm-Šidák correction for multiple testing. For (**B**, **C**, **E**, **F**) p-values are from Mantel Cox test. In (**G**) p-values are from Log rank test in cBioPortal and q-values (also from cBioPortal) employ Benjamini Hochberg procedure to correct for false discovery. Source data are provided as a source data file.

Positive controls AZD7648 and ART558 showed expected effects on NHEJ (Fig. 4A), HR (Fig. 4B), and MMEJ (Fig. 4C), as in Fig. 2D. Conversely, none of the negative control inhibitors showed major effects on any pathway. Importantly, three molecules strongly inhibited both HR and MMEJ: AZD1390 (ATM inhibitor, ATMi), Birabresib (BET inhibitor), and Panobinostat (HDAC inhibitor), (Fig. 4B, C, see Fig. S8 for representative flow cytometry plots). Similar results were obtained in SF295 cells (Fig. 4D). These drugs did not affect the cell cycle distribution of U251 cells treated for 24 h (Fig S9), suggesting that HR/MMEJ inhibition is not an artifact of cell cycle perturbation.

We next used genomically-integrated reporter cell lines, U251 DR-GFP and U251 EJ2-GFP[40], to assess the hits identified by FM-DSBR. Transfection with SceI expression vector forms a DSB at the reporter locus, and DSBR by either HR (DR-GFP cells) or MMEJ (EJ2-GFP) causes GFP fluorescence. Positive control inhibitors confirmed the fidelity of each reporter cell line: the HR inhibitor, BO2[41], suppressed expression of the HR reporter, DR-GFP (Fig. 4E), and increased expression of EJ2-GFP (Fig. 4F). Conversely, MMEJ inhibitors ART558 and NVB inhibited expression of the MMEJ reporter, EJ2-GFP. Finally, the DNA-PKcs inhibitor, AZD7648, increased expression of both DR-GFP and EJ2-GFP, consistent with our results using the FM-DSBR assay (Fig. 4B, C).

The ATMi, AZD1390, inhibited both DR-GFP (Fig. 4E) and EJ2-GFP (Fig. 4F, see Figs. S10 and S11 for representative flow cytometry plots), as did an additional ATMi, KU60019 (Fig. S12). Panobinostat also inhibited DR-GFP and EJ2-GFP; however, we note that it was toxic at doses above 10 nM under the 72 h treatment time used for this assay. Conversely, Birabresib inhibited DR-GFP but not EJ2-GFP. Unlike the other inhibitors tested, Birabresib caused robust SceI-independent GFP-fluorescence in EJ2-GFP cells (Fig. S10), confounding our ability to assess its effects on EJ2-GFP MMEJ. We note that BETi causes transcription-replication collisions[42]. Unlike our plasmid-based reporters, genomic DSBR reporters are replicated when cells divide and may be prone to DSB formation during BETi treatment. Nonetheless, these

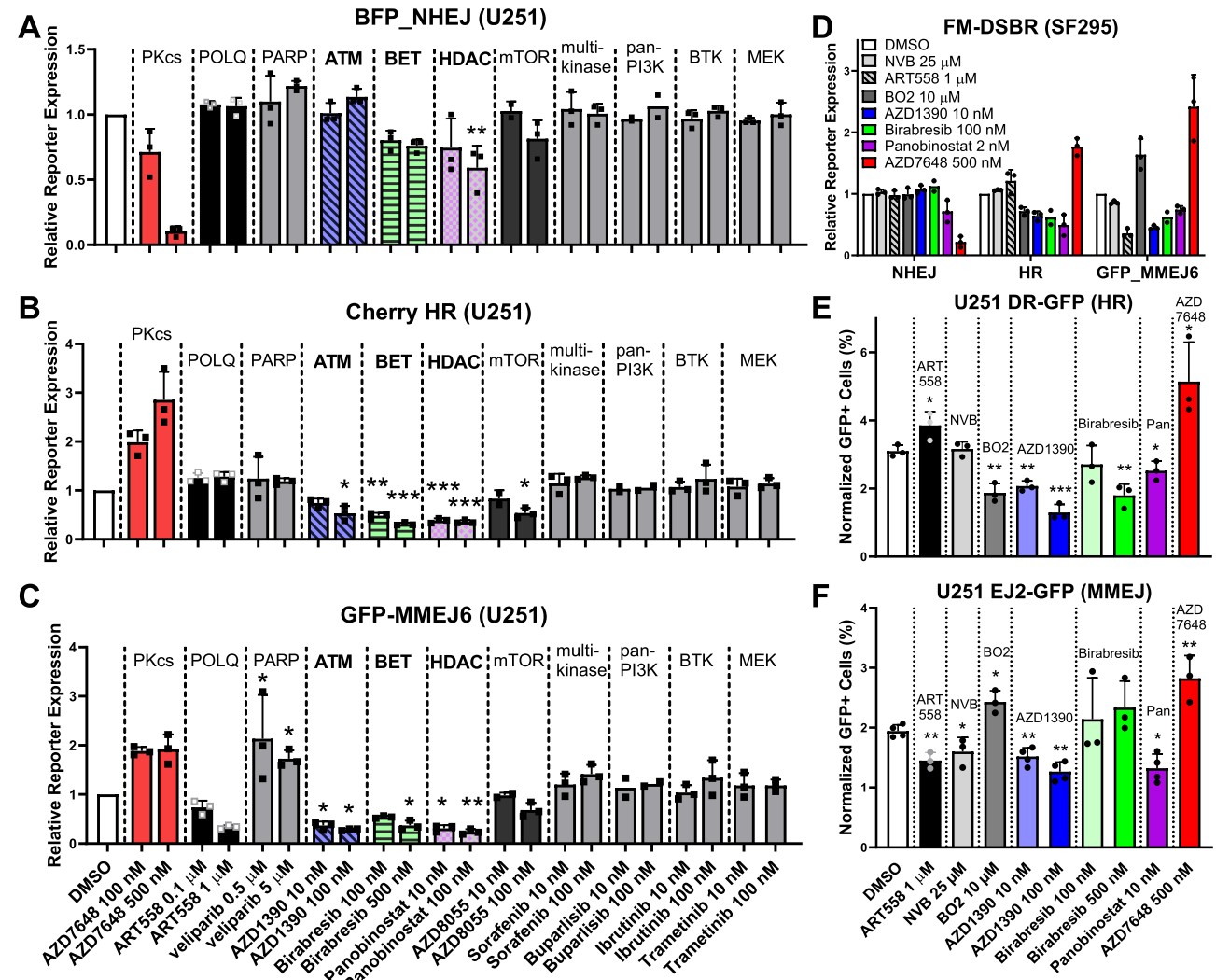

**Fig. 4 | Fluorescence-based multiplexed double strand break repair (FM-DSBR) analysis identifies dual HR/MMEJ inhibitors in GBM. A–C** FM-DSBR in U251 cells treated with indicated inhibitor for 2 h and transfected with BFP_NHEJ, Cherry_HR, GFP_MMEJ6, and pMax Am Cyan plasmid as an internal control for transfection efficiency (n = 3 independent experiments except Buparlisib where n = 2). Cells were analyzed 24 h after transfection by flow cytometry and reporter expression was calculated as previously described[24] and normalized to DMSO control. The target of each inhibitor is displayed above and dashed lines separate columns representing data for different inhibitors. See Fig. S8 for representative flow cytometry data and gating scheme. **D** FM-DSBR in SF295 under the same conditions as U251. **E, F** Analysis of HR by DR-GFP reporter and MMEJ by EJ2-GFP reporter in U251 reporter cell lines[40]. Cells were treated with drug, immediately transfected with pCBASceI plasmid and pMax BFP transfection control and analyzed after 72 h. Data are presented as the mean of 3 independent experiments, except for (**F**) where n = 4 for DMSO, AZD1390, and Panobinostat. Error bars show SD. In (**A–C**), statistical comparison was to DMSO by one-way ANOVA with Dunnett's multiple comparisons test. In (**E, F**) statistical comparison was to DMSO using unpaired two-tailed t-test. *: p < 0.05, **: p < 0.01, ***: p < 0.001. Source data and exact p-values are available in the source data file.

data from DR-GFP and EJ2-GFP cells indicate that the inhibitors identified by our plasmid-based FM-DSBR screen also suppress HR and MMEJ of genomic DSBs.

### AZD1390 inhibits HR and MMEJ and potentiates efficacy of DNA-damaging therapy in GBM

Having identified three dual HR/MMEJ inhibitors, we tested each drug in combination with TMZ or radiation in U251 cells. Importantly, we also tested the MMEJ inhibitor, NVB, and the HR inhibitor, BO2. NVB did not sensitize U251 to radiation (Fig. 5A) or TMZ (Fig. 5B), while BO2 significantly sensitized to both agents, although this effect was modest. Importantly, AZD1390, Birabresib, and Panobinostat potentiated both radiation and TMZ with greater potency than NVB or BO2, suggesting that dual HR/MMEJ inhibition is superior to inhibition of either individual pathway. Consistent with this interpretation, ART558/BO2 or NVB/BO2 combination were

superior to either individual agent in potentiating killing by TMZ in U251 cells in a viability assay (Fig. S13).

Although AZD1390, Birabresib, and Panobinostat inhibited HR and MMEJ, Birabresib and Panobinostat were less efficacious in potentiating TMZ. We hypothesized that these inhibitors affect other pathways involved in TMZ resistance and employed FM-HCR assays specific for MMR (mOrange_G:G) and MGMT ($O^6$-MeG_m-Plum). Indeed, we found that Birabresib significantly suppressed MMR activity, Panobinostat significantly enhanced activity of MGMT, and AZD1390 had no significant effect on either pathway (Fig. 5C). Since suppression of MMR and increased MGMT activity are established TMZ resistance mechanisms, we hypothesize that these changes weaken the effectiveness of BETi and HDACi in combination with TMZ. Taken together, our data support AZD1390 as the most promising drug for TMZ combination therapy, so we subjected it to further analysis.

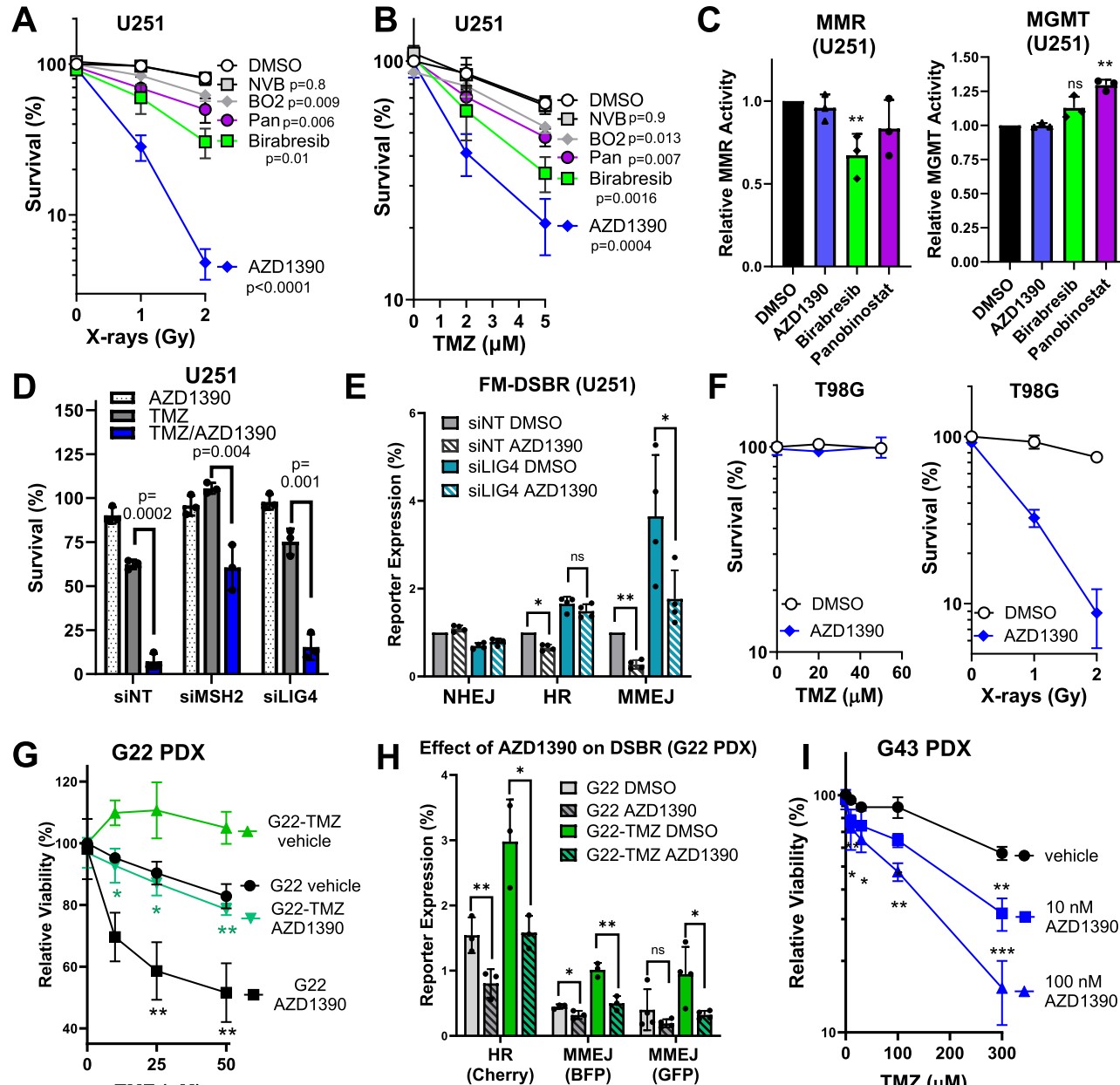

**Fig. 5 | AZD1390 inhibits HR and MMEJ and potentiates cell killing by DNA-damaging therapy in treatment-naïve and treatment resistant GBM cells.**
**A, B** Clonogenic survival of U251 treated with TMZ or radiation in combination with: DMSO control, AZD1390 (10 nM), Birabresib (50 nM), Panobinostat, abbreviated "Pan" (2 nM), novobiocin (NVB, 25 μM), or BO2 (5 μM). **C** Analysis of MMR and MGMT activity in U251 treated with 100 nM AZD1390, Birabresib, or Panobinostat for 2 h and transfected with mOrange_G:G (MMR) and mPlum_$O^6$-MeG (MGMT) with pMax_BFP transfection control. **D** Clonogenic survival of U251 transfected with indicated siRNA, treated with DMSO, TMZ, or TMZ/AZD1390 for 96 h. **E** FM-DSBR of U251 transfected with indicated siRNAs. **F** Clonogenic survival of T98G treated with AZD1390 (10 nM) in combination with radiation or TMZ (n = 2). **G** Relative viability of G22 and G22-TMZ treated (120 h) with vehicle (DMSO), AZD1390 (50 nM), TMZ, or combination. **H** FM-DSBR in G22 or G22-TMZ following 1 h DMSO or AZD1390 (50 nM) treatment. **I** Relative viability of G43, same as (**F**). Data are represented as the mean of three independent experiments and error bars show SD. Statistical comparisons are by unpaired two-tailed t-test. In (**A**, **B**, **G**, **I**), p-values are from comparison to vehicle control. For (**A**, **B**), the p-value is displayed for only the high dose. In (**G**), p-values are below each point and in (**I**), p-values are above the point. Holm-Šidák correction for multiple testing was applied in (**D**, **E**, **G–I**). Source data are provided as a source data file.

We first knocked out ATM, which ablated the treatment-enhancing effects of AZD1390 as well as its inhibition of HR/MMEJ (Fig. S14), confirming on-target activity of this inhibitor. We then tested whether AZD1390 could enhance TMZ therapy in cells with defined resistance mechanisms induced by genetic knockdown. U251 cells transfected with non-targeting siRNA (siNT) were sensitive to TMZ, and co-treatment with AZD1390 markedly decreased survival (Fig. 5D). Knockdown of MSH2, resulting in a 60% decrease in MMR activity (Fig. S15), rendered cells completely resistant to TMZ, consistent with

previous reports that even a partial loss of MMR imparts substantial TMZ resistance in GBM cells[19]. Interestingly, AZD1390 significantly sensitized MSH2-depleted cells to TMZ (Fig. 5D), suggesting that GBMs that acquire TMZ resistance by partial loss of MMR can be targeted by this combination. We also depleted LIG4, which imparted slight TMZ resistance compared to siNT (Fig. 5D). TMZ/AZD1390 combination was slightly less effective in siLIG4 cells, possibly due to enhanced HR/MMEJ in these cells (Fig. 5E). To test whether ATM inhibition potentiates killing by TMZ through mechanisms other than inhibiting repair of

$O^6$-MeG-induced DSBs, we employed the MGMT-proficient cell line, T98G. We found that AZD1390 markedly potentiated cell killing by radiation but had no effect for TMZ (Fig. 5F), indicating that the combination is not highly toxic to cells capable of repairing $O^6$-MeG lesions. Taken together, these data suggest that AZD1390 sensitizes GBM cells to DNA-damaging agents only when DSBs are formed.

Finally, we employed AZD1390/TMZ combination in GBM PDX samples. AZD1390 enhanced killing by TMZ in treatment naïve G22 (Fig. 5G), where it also significantly suppressed HR/MMEJ (Fig. 5G). Importantly, G22-TMZ, which was completely resistant to TMZ under these conditions, was partially re-sensitized by AZD1390. Finally, we assessed TMZ/AZD1390 in G43, an MGMT-proficient GBM that responds only weakly to high-dose TMZ regimens[43]. Strikingly, AZD1390 (10 nM or 100 nM) potentiated killing of G43 cells by TMZ at doses ranging from 10 to 300 μM (Fig. 5I). Taken together, our data indicate that AZD1390 enhances DNA-damaging therapy by inhibiting HR/MMEJ and can enhance cell killing by TMZ in both treatment-naïve and treatment-resistant cells.

## AZD1390 suppresses end resection and radiation-induced phosphorylation of DSB end protection factors

We next investigated the mechanism by which AZD1390 inhibits HR/MMEJ in GBM. We hypothesized that AZD1390 inhibits DSB end resection, so we used a previously reported DSB resection assay[44]. Briefly, cells are treated with camptothecin (CPT), fixed/permeabilized, and stained with an antibody against replication protein A (RPA), which coats ssDNA exposed by nucleolytic resection of DSBs. Treatment of U251 cells with CPT induced robust RPA staining that was suppressed by AZD1390 (1 h pre-treatment before addition of CPT) in a dose-dependent manner (Fig. 6A, B). X-irradiation with 10 Gy induced a less pronounced RPA signal that was also strongly suppressed by AZD1390 pre-treatment. Similarly, siRNA knockdown of ATM suppressed CPT- and radiation-induced RPA staining (Fig. S16), although to a lesser degree than AZD1390. We gated cells by DNA content and found that CPT mainly induced RPA staining in S and G2 phase (Fig. 6C). AZD1390 markedly suppressed CPT-induced RPA signal in S phase but showed a lesser inhibitory effect in G2 phase. Conversely, we observed pronounced radiation-induced RPA staining in G2 that was almost completely inhibited by AZD1390 (Fig. 6D), consistent with previous results that ATM is required for HR of radiation-induced breaks in G2[9]. We conclude that AZD1390 suppresses DSB end resection in GBM, especially in S and G2 phase.

ATM promotes HR via multiple mechanisms including activating DSB resection and promoting HR completion after Rad51 nucleofilament formation[45–47], but a clear picture of ATM's role in MMEJ is lacking. We considered emerging evidence that MMEJ functions not only in repair of S phase DSBs but also those that persist until mitosis[48,49]. To determine whether ATM plays a cell cycle-specific role in DSBR, we arrested U251 cells in G1, S, or M (treatment with palbociclib, aphidicolin, or nocodazole, respectively), treated with vehicle or AZD1390, and measured FM-DSBR. AZD1390 had little effect on NHEJ (Fig. 6E) in any cell cycle phase. Conversely, HR activity, which was relatively low in G1 and M and highest in S, was suppressed by AZD1390, especially in S phase. Interestingly, we observed a cell cycle-specific effect of AZD1390 on MMEJ. Baseline MMEJ activity was markedly lower in G1-arrested cells and was unaffected by AZD1390. Conversely, arrest in either S or M phase was associated with higher MMEJ activity that was suppressed by AZD1390. We conclude that AZD1390 suppresses S-phase HR and MMEJ along with mitotic MMEJ but does not suppress MMEJ in G1.

We investigated the mechanism of HR/MMEJ inhibition by AZD1390 using siRNA-mediated knockdown of Mre11 and CtIP—which initiate DSB end resection—or BLM, which functions in long-range resection[11]. All three knockdowns suppressed mCherry_HR expression,

and this was further suppressed by AZD1390 (Fig. 6F), consistent with a role for ATM in promoting HR by additional mechanisms aside from initiating end resection[47]. MMEJ was similarly suppressed by knockdown of resection proteins and inhibited further by AZD1390, although this was statistically significant only in si*RBBP8* cells (Fig. 6G). These data are consistent with a role for ATM in multiple steps of HR and suggest that in addition to initiating resection, ATM may have other roles in MMEJ.

To further investigate this, we conducted an unbiased phosphoproteomics screen to identify ATM targets. We treated SF295 cells with vehicle or AZD1390 (100 nM) for 1 h and then irradiated with 6 Gy or mock (no irradiation). We collected cell pellets 1 h after irradiation, repeated this three times on separate days, lysed the pellets, multiplexed samples for quantitative phosphoproteomics using tandem mass tags (TMT), and analyzed by liquid chromatography-mass spectrometry. We found 505 phosphorylation sites (phospho-sites) that were significantly different (p < 0.05) between 6 Gy DMSO and 6 Gy AZD1390 (Fig. 6H) and 404 significant sites between 0 Gy DMSO and 6 Gy DMSO (Fig. S17, Table S3). We considered only the phospho-sites that were significantly more abundant in the 6 Gy condition compared to both 0 Gy and 6 Gy AZD1390. These represent radiation-induced phospho-sites that are suppressed by AZD1390 and included the ATM autophosphorylation site, S2296 (Fig. 6I), previously reported ATM phospho-sites on the pro-resection proteins, nibrin (NBN) S343, and Rad50 S635[50,51], and additional sites on nibrin (S615 and S397) and Rad50 (S690). These data are consistent with a role for ATM in activating DSB end resection by phosphorylating resection proteins. Interestingly, we also detected phospho-sites on three end protection proteins (Fig. 6J), 53BP1 (9 phospho-sites), Rif1 (3 phospho-sites), and Mettl16 (S419 and S463). Phospho-sites on 53BP1 and Rif1 remain to be characterized at a molecular level, but phosphorylation of Mettl16 S419 releases Mre11 from sequestration by an RNA-protein complex, allowing it to initiate end resection and promote HR/MMEJ[52]. Taken together, these data suggest that, in GBM, ATM controls DSBR end resection by multiple mechanisms, including activating pro-resection proteins such as nibrin and Rad50 and by deactivating end protection proteins such as Mettl16.

## AZD1390 inhibits HR/MMEJ in *TP53*-mutant GBMs

ATMi potentiate killing by radiation and doxorubicin in *TP53*-mutant cancer cells but not those that are *TP53*-wild type (WT)[6,53,54]. To determine whether the efficacy of TMZ/AZD1390 combination is also restricted by *TP53* mutational status, we assessed this combination in additional GBM cell lines. Similar to results in clonogenic assays, AZD1390 markedly potentiated killing by TMZ in U251 (*TP53*-mutant, MGMT-null) but not T98G (*TP53*-mutant, MGMT-proficient, Fig. 7A). Strikingly, AZD1390 had no effect in U87 or A172, both of which are MGMT-null but *TP53*-WT (Fig. 7A), indicating that AZD enhances killing by TMZ only in GBMs with *TP53* mutations.

To investigate the basis for this observation, we employed FM-DSBR in 12 GBM cell lines and xenografts. Interestingly, AZD1390 inhibited HR (Fig. 7B) and MMEJ (Fig. 7C) in all six *TP53*-mutant GBMs. By contrast, AZD1390 inhibited HR in three of six *TP53*-WT GBMs and significantly inhibited MMEJ in only one. We expanded this analysis to additional cancer cell lines, where we saw robust dual HR/MMEJ inhibition by AZD1390 or the ATMi, KU60019, in *TP53*-mutant GBM, breast, and skin cancer cells but lesser inhibition in most *TP53*-WT cancer cell lines (Fig. S18). We conclude that ATMi suppress HR and MMEJ more potently in *TP53*-mutant cancers.

Mutation of p53 enhances HR activity in multiple cell types[55], and POLQ expression is upregulated in *TP53*-mutant cancers, including GBM[56]. However, the full spectrum of DSBR in *TP53*-WT vs. *TP53*-mutant GBM is unknown. We compared NHEJ, HR, and MMEJ activity (Fig. 7D) in the same panel of GBM cell lines and xenografts

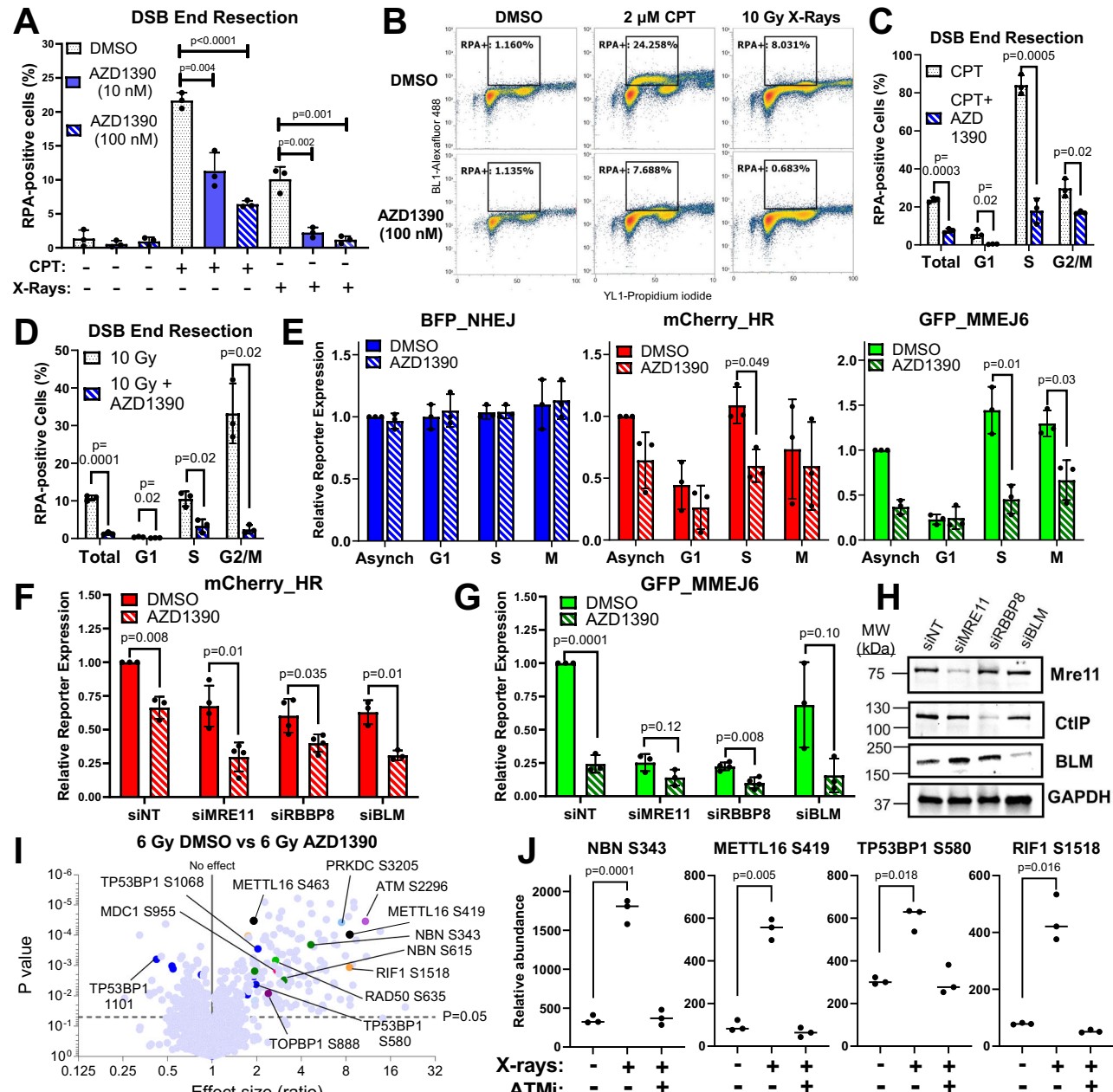

**Fig. 6 | AZD1390 suppresses DSB end resection and radiation-induced phosphorylation of DSB end protection factors. A** RPA staining in U251 pre-treated with DMSO or AZD1390 (1 h) followed by DMSO, 2 μM camptothecin (CPT), or 10 Gy X-rays and collected 1 h later. **B** Representative flow cytometry plots from (**A**). **C**, **D** RPA staining data from (**A**) as a function of cell cycle phase, as determined by DNA content (PI staining). **E** NHEJ, HR, and MMEJ in asynchronous (Asynch) U251, or U251 arrested with palbociclib (G1), aphidicolin (S), or nocodazole (mitosis, M) for 18 h prior to AZD1390 treatment (100 nM, 1 h) and FM-DSBR. **F**, **G** HR and MMEJ activity in U251 transfected with indicated siRNA 72 h prior to treatment with DMSO or AZD1390 (100 nM, 1 h) and transfected with HR/MMEJ reporters. **H** Western blot

for experiments in (**F**, **G**). The samples derive from the same experiment; however, the blots for Mre11 and CtIP are from the same gel, while BLM and GAPDH are from another gel. **I** Volcano plot of phosphorylated peptides detected by LC-MS/MS in SF295 cells treated with DMSO or AZD1390 (100 nM) for 1 h and then treated with 6 Gy (n = 3 biologically independent experiments). The x-axis represents the fold-difference in expression, with higher values representing greater abundance in the DMSO control. **J** Selected phosphorylated peptides from (**G**). In (**A**, **C**–**G**, **F**), data are presented as the mean of three independent experiments, error bars show SD, and p-values are from unpaired two-tailed t-test. Holm-Šídák correction for multiple testing was applied in (**C**–**G**). Source data are provided as a source data file.

used in Fig. 7B, C. The average NHEJ efficiency was similar between *TP53*-WT and *TP53*-mutant (mean: 10.6% vs 11.4%, median: 11.2% vs. 13%). Conversely, HR efficiency was ~2-fold higher in *TP53*-mutant vs. *TP53*-WT (mean: 1.4% vs. 3.2%, median: 1.4% vs. 2.7%), although this difference did not reach statistical significance (p = 0.055). MMEJ efficiency showed the largest difference between *TP53*-WT and *TP53*-mutant, with the latter exhibiting a median MMEJ activity that was nearly 6-fold higher than wild type (mean: 0.09% vs. 0.34%, median:

0.059% vs. 0.275%, p = 0.025). We conclude that, compared to *TP53*-WT GBMs, those with *TP53* mutation display heightened HR and MMEJ.

We extended this analysis of ATM-dependent DSBR to GBM patients by analyzing gene expression in GBMs from the TCGA Pan-Cancer Atlas dataset using cBioportal[31–33]. We found 2239 genes expressed more highly (q < 0.05 in cBioPortal) in *TP53*-mutant GBMs than in *TP53*-WT and subjected these genes to pathway analysis using

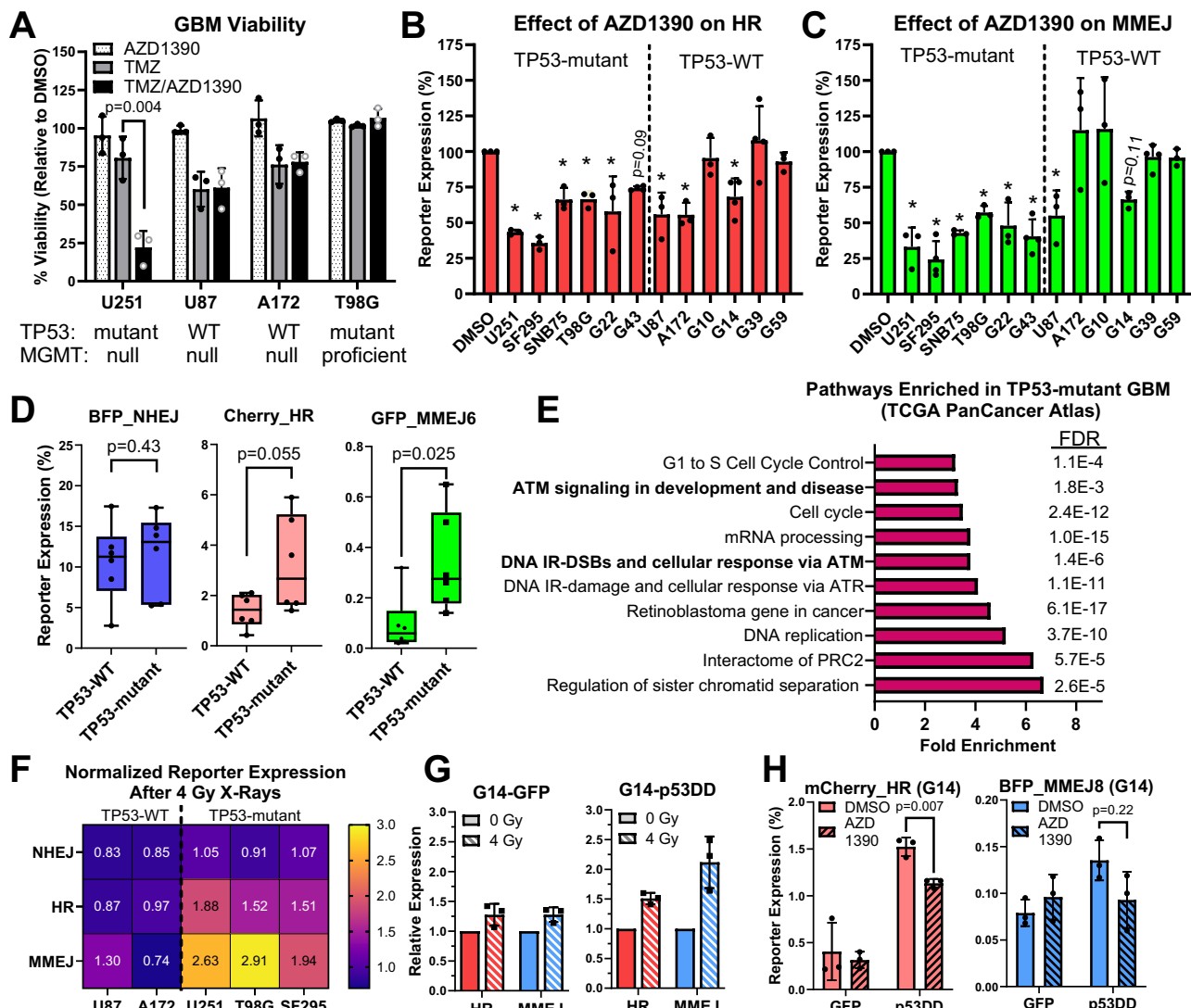

**Fig. 7 | AZD1390 suppresses elevated ATM-dependent HR and MMEJ in *TP53*-mutant GBM. A** Viability of GBM lines treated with TMZ (25 µM), AZD1390 (10 nM), or both for 6 days and normalized to vehicle (PBS). WT: wild type. **B**, **C** Cherry_HR and GFP_MMEJ6 expression (normalized to DMSO control) for cells treated with AZD1390 (1 h, 100 nM) before transfection. **D** Box and whisker plot showing NHEJ, HR, and MMEJ activity in *TP53*-WT or *TP53*-mutant GBMs (n = 6 GBM lines per group). The box displays the 25–75th percentile, the median is indicated by a solid line, and the whiskers (error bars) display the range (min. and max.). **E** Pathway analysis of genes upregulated in *TP53*-mutant GBMs. The top ten pathways are ranked by fold enrichment, false discovery rate (FDR) is displayed for each. **F** Heatmap showing NHEJ, HR, and MMEJ activity in the indicated cell line after pre-

irradiation with 4 Gy X-rays. Data are reported as fold-change relative to 0 Gy control according to the color scale at right with the mean (n = 3) displayed in each cell. **G** mCherry HR and BFP_MMEJ8 activity in G14-GFP or G14-p53DD following pre-irradiation with 0 Gy or 4 Gy. **H** mCherry_HR and BFP_MMEJ8 activity in G14 cells stably expressing GFP or dominant negative p53DD (residues 300–393). Cells were treated with DMSO or AZD1390 (100 nM) and transfected 1 h later. Data are presented as the mean of three independent experiments, error bars show SD in all panels except for (**D**) where they show the range. In (**A, D, H**), p-values are from unpaired two-tailed t-test. In (**B, C**), p-values are from one-way ANOVA with Dunnett's test for multiple comparisons. * indicates p < 0.05. Exact p-values for (**B, C**) and all source data are available in the source data file.

DAVID bioinformatics resource. Strikingly, both ATM signaling and cellular response to DSBs via ATM were among the top 10 significantly (FDR < 0.01) enriched pathways in *TP53*-mutant GBM (Fig. 7E). Coupled with stronger response of HR/MMEJ to AZD1390 in *TP53*-mutant GBM (Fig. 7B, C), this suggests an increased reliance on ATM-dependent HR and MMEJ in response to DNA damage. We further explored this hypothesis by analyzing DSBR in response to genomic DNA damage in a subset of GBM cells. We irradiated cells with 4 Gy X-rays—chosen due to the ability to rapidly induce DSBs—and transfected FM-DSBR reporters 1 h later. *TP53*-mutant U251, SF295, and T98G cells showed a robust increase in HR and MMEJ immediately after 4 Gy irradiation (Fig. 7F), and this increase was suppressed by AZD1390 (Fig. S19). Interestingly, HR/MMEJ were not significantly elevated after irradiation

in U87 and A172, suggesting that *TP53* mutant GBMs undergo a more robust activation of ATM-dependent HR/MMEJ following genomic DNA damage.

The GBM cell lines compared in this study originated in different individuals and are thus genetically distinct. To directly compare the effect of p53 loss on DSBR, we created isogenic pairs using *TP53*-WT GBM cells stably expressing control vector (GFP) or a C-terminal p53 fragment (residues 300-393, p53DD) that inactivates WT p53 by forming non-functional oligomers[57]. We assessed DSBR after irradiation in G14-GFP and G14-p53DD and found that, while HR/MMEJ were only modestly enhanced after 4 Gy in G14-GFP (28% increase for HR and MMEJ), both pathways were markedly enhanced in G14-p53DD (50% increase in HR and 110% increase in MMEJ, Fig. 7G).

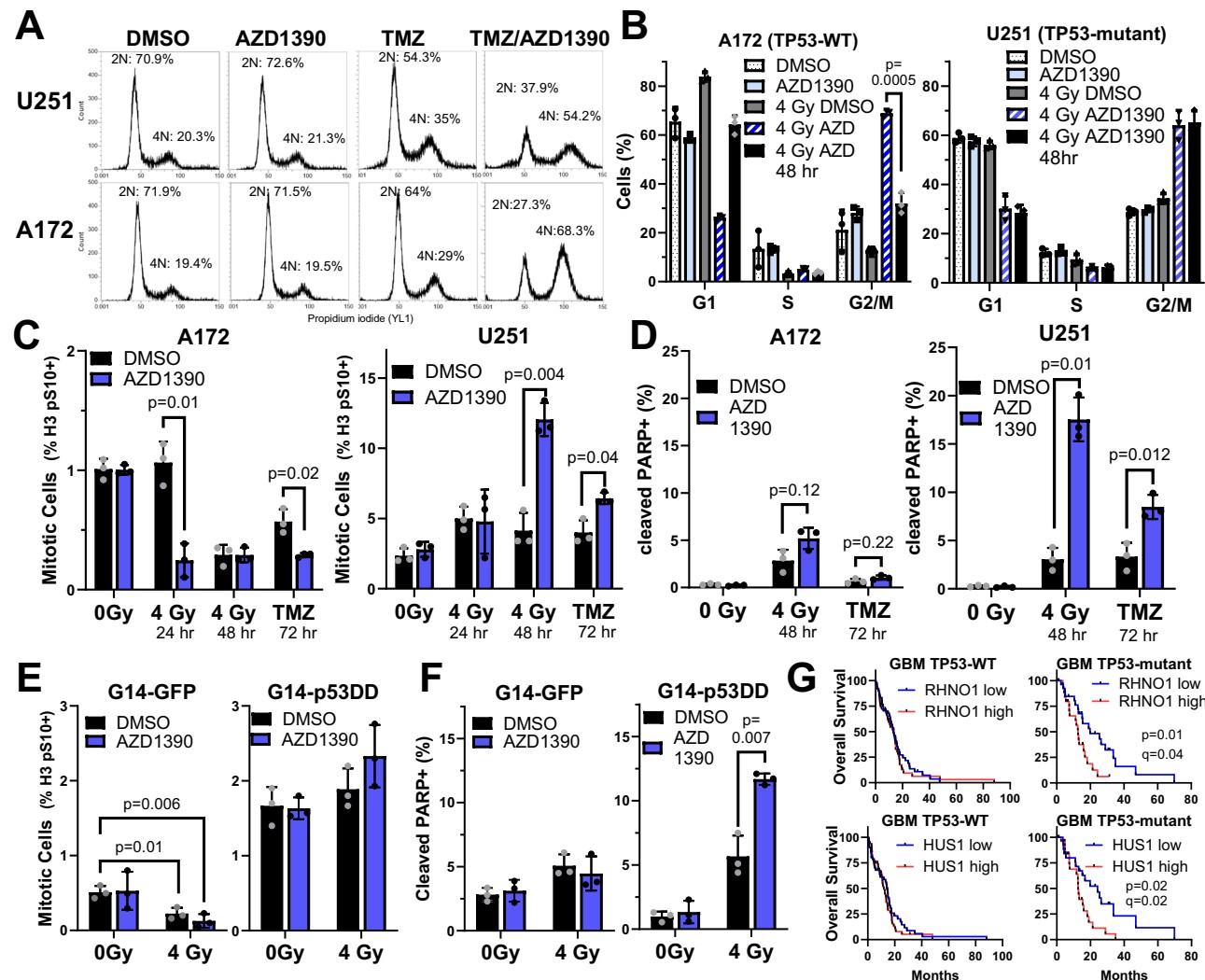

**Fig. 8 | A dysfunctional G2/M checkpoint renders *TP53*-mutant GBMs sensitive to ATM inhibition. A** Cell cycle profiles of U251 or A172 treated with DMSO (0.1%), AZD1390 (10 nM), TMZ (10 μM) or TMZ/AZD1390 for 72 h, fixed, and stained with propidium iodide. Experiment was repeated thrice with similar results. **B** Cell cycle distribution of U251 and A172 after the indicated treatment for 24 h (first four bars) or 48 h (last bar). **C** Immunostaining of U251 or A172 cells with anti-phospho S10 H3 antibody after indicated treatment in the presence or absence of AZD1390 (10 nM) and (**D**) Immunostaining with antibody against cleaved PARP (Asp214) 48 h after treatment. Immunostaining of G14-GFP and G14-p53DD cells 24 h after indicated treatment using anti-phospho S10 H3 (**E**) or anti-cleaved PARP Asp214 (**F**). **G** Survival analysis of TP53-WT (n = 99) or TP53-mutant (n = 51) GBM patients stratified by median expression of RHNO1 or HUS1 and compared using cBioportal, p-values from Log rank test with q-values employing Benjamini Hochberg procedure to correct for false discovery. In (**B**–**F**), data are presented as the mean of three independent experiments, error bars show SD, and p-values are from unpaired two-tailed t-test with Holm-Sidak correction for multiple comparisons. Source data are provided as a source data file.

Finally, we explored whether p53 loss alters the baseline efficiency and ATM-dependency of HR and MMEJ. In G14, p53DD expression increased HR efficiency by nearly 4-fold (1.5% vs. 0.4%) and MMEJ by nearly two-fold (0.14% vs. 0.08%, Fig. 7H). Similar effects were seen in an additional isogenic pair, G10-GFP and G10-p53DD (Fig. S19). AZD1390 had little effect on HR or MMEJ in G14-GFP cells, but importantly, had greater inhibitory effects in G14-p53DD, especially for HR. Similar results were obtained for G10-GFP/G10-p53DD pair (Fig. S19). This suggests that loss of p53 function at least partially explains the enhanced HR/MMEJ activity and increased usage of HR/MMEJ in *TP53*-mutant GBMs.

**A defective G2/M checkpoint renders *TP53*-mutant GBMs sensitive to AZD1390 in combination with DNA-damaging therapy**

In *TP53*-WT GBMs including U87 and A172, AZD1390 inhibits HR but does not enhance killing by TMZ (Fig. 7A) or radiation (ref.)[6]. p53 enforces G1 and G2/M cell cycle checkpoints after DNA damage,

protecting cells from the toxic effects of attempting to replicate or divide without repairing DSBs[58]. We hypothesized that intact cell cycle checkpoints protect *TP53*-WT GBMs from cell death when DSBs remain repaired due to ATMi. To test this, we first compared the cell cycle distribution of A172 (*TP53*-WT) or U251 (*TP53*-mutant) cells treated with TMZ or radiation in the presence or absence of AZD1390.

TMZ treatment for 72 h slightly increased the proportion of cells with 4N DNA content (G2 or M phase) in both U251 and A172 (Fig. 8A). This was markedly increased by AZD1390 co-treatment in both cell lines, suggesting that AZD1390 causes TMZ-induced breaks to persist into G2 and trigger G2 arrest in U251 and A172. Slightly different results were obtained for radiation: 24 h after irradiation with 4 Gy, A172 showed a small but significant increase in G1 phase cells while U251 did not (Fig. 8B), consistent with p53-dependent G1 arrest only in *TP53*-WT A172. Interestingly, 4 Gy/AZD1390 combination caused marked G2/M accumulation in both A172 and U251. This suggests that, although A172 initiates G1 arrest after DNA damage, this is blocked by

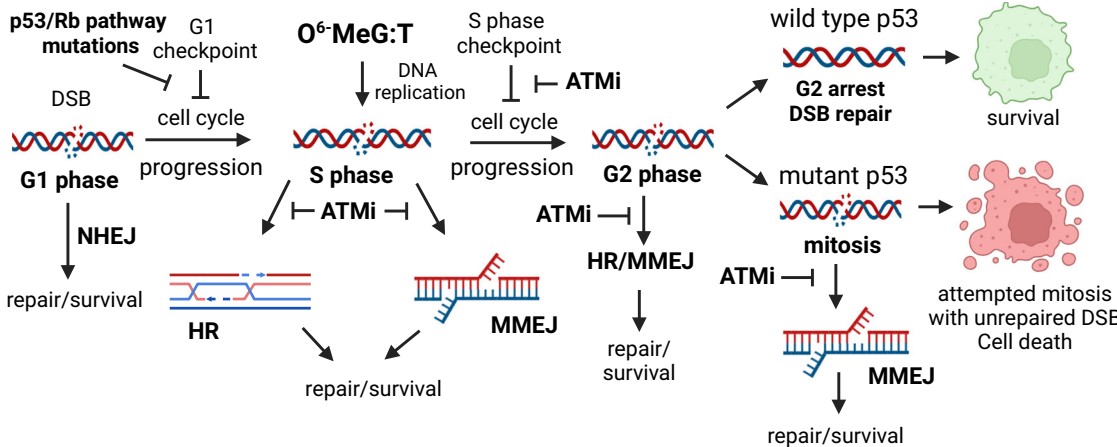

**Fig. 9 | Model for effect of ATMi on GBM cells following formation of a DSB by radiation or TMZ.** Cell cycle progression is shown from left to right. Model created using Biorender.

AZD1390, consistent with previous reports that the G1 checkpoint is p53-dependent, initiated by ATM, and blocked by ATMi[59,60]. Taken together, our data suggest that combination of ATMi with DNA-damaging therapy causes DSBs to accumulate in G2 phase regardless of *TP53* status.

Interestingly, 48 h after 4 Gy/AZD1390 combination, U251 cells continued to exhibit 4N DNA content (G2 or M phase), while A172 had recovered and apparently progressed through mitosis into G1 phase (Fig. 8B, 4 Gy AZD1390 48 h). DNA content alone cannot distinguish G2 cells from mitotic cells, so we stained for the mitotic marker, phosphorylation of S10 on histone H3 (H3 pS10). Strikingly, we found that U251 cells continued to enter mitosis after 4 Gy irradiation or 10 µM TMZ, while A172 cells displayed a marked reduction in mitotic cells after either treatment (Fig. 8C). Combination with AZD1390 markedly enhanced the proportion of mitotic U251 cells after TMZ or 4 Gy but reduced the proportion of mitotic A172 cells (Fig. 8C). These data suggest that, while A172 cells undergo G2 arrest and repair their DSBs prior to mitosis, U251 cells do not and instead enter M phase with unrepaired DSBs and undergo cell death. Consistent with such an interpretation, 4 Gy/AZD1390 or TMZ/AZD1390 caused a marked increase in staining for cleaved PARP, a marker for apoptotic cell death, in U251 (Fig. 8D). A172 cells showed minimal staining for cleaved PARP when AZD1390 was combined with radiation or TMZ, suggesting that these cells arrest in G2 long enough to repair DSBs, enabling successful mitosis.

Finally, we extended our analysis to an isogenic GBM pair: G14-GFP and G14-p53DD. Similarly to U251, G14-p53DD continued to enter mitosis 24 h after 4 Gy or 4 Gy/AZD1390 (Fig. 8E), while G14-GFP showed a marked decrease in mitotic cells 24 h after 4 Gy or 4 Gy/AZD1390 (Fig. 8E). Finally, AZD1390 treatment enhanced the proportion of G14-p53DD cells staining positive for cleaved PARP 48 h after 4 Gy irradiation (Fig. 8F), suggesting that like *TP53*-mutant U251, these cells fail to progress through mitosis due to unrepaired DSBs and eventually undergo cell death. Importantly, in G14-GFP, treatment with AZD1390 before irradiation with 4 Gy did not significantly affect the proportion of cleaved PARP-positive cells, similar to results in *TP53*-WT A172. These data suggest that the G2/M checkpoint protects *TP53*-WT GBMs from mitotic entry and cell death following DNA damage/ATMi combination, while a defective G2/M checkpoint allows *TP53*-mutant GBMs to enter mitosis, which they fail to complete due to unrepaired DSBs, ultimately resulting in cell death. Recent evidence suggests that MMEJ is activated by Plk1, RHINO, and HUS1 in mitosis, where it represents a final failsafe for repairing DSBs that persist until mitosis[48,49]. Interestingly, expression of *RHNO1* (which encodes RHINO) and *HUS1* were negatively correlated with survival in *TP53*-

mutant GBM but not *TP53*-WT (Fig. 8G), suggesting that checkpoint defects in *TP53*-mutant GBMs render them reliant on mitotic MMEJ to survive treatment.

Taken together, our data are consistent with a model where ATMi block repair of DSBs in S and G2 phase by inhibiting HR/MMEJ (Fig. 9). These DSBs can be formed directly by radiation in any phase of the cell cycle or in S phase by replication of TMZ-induced $O^6$-MeG: T mispairs. DSBR inhibition coupled with inhibition of S phase arrest by ATMi causes an accumulation of unrepaired DSBs in G2 phase, triggering G2 arrest. *TP53*-WT GBMs sustain this arrest and repair their DSBs prior to mitosis, which they complete successfully and re-enter the cell cycle. Conversely, *TP53*-mutant GBMs do not sustain G2 arrest and continue to enter mitosis with unrepaired DSBs, where inhibition of mitotic MMEJ by ATMi prevents the final attempt at repair, leading to cell death.

## Discussion

Investigating the molecular mechanisms of chemotherapy resistance has yielded novel treatment strategies and improved our understanding of DSBR[5,7,15]. Genomically integrated DSBR reporters are an invaluable tool in these investigations, but these reporters have been difficult to multiplex and are limited to a small number of genetically engineered cell lines. Although recent advances have expanded the ability to multiplex DSBR measurements[29,61], it has remained challenging to assess DSBR in large numbers of cell lines and primary samples. Our plasmid-based approach overcomes some of these limitations, allowing us to investigate DSBR mechanisms and treatment responses across many cell lines and patient-derived samples.

We identified DSBR alterations in TMZ-resistant GBM xenografts, namely deficient NHEJ coupled with enhanced HR/MMEJ. Additionally, we found that GBMs with genetic signatures of increased end resection (low SHLD1 or high TRIP13) have poor prognosis when receiving TMZ. Consistent with an important role for resection-dependent pathways (HR/MMEJ) in TMZ resistance, knockdown of *RBBP8* enhanced killing by TMZ, while knockdown of *LIG4*, which stimulates resection-dependent repair, imparted weak TMZ resistance (Fig. 1K). Interestingly, *POLQ* knockdown slightly enhanced killing by TMZ, but the POLQi ART558 and NVB were ineffective in enhancing killing by TMZ. The basis for this disparity is unclear; however, gap-filling by Pol theta may play an important role in chemoresistance[62], and additional work is needed to determine how POLQi affect gap-filling and MMEJ, particularly because NVB and ART558 act upon different domains of the polymerase. We also found that the HR inhibitor, BO2, was more effective than ART558 or NVB in potentiating killing by TMZ (Figs. 5A and S13), suggesting that HR is the predominant pathway for

repair of TMZ-induced DSBs. However, the ability of either HR or MMEJ to act in S and G2 phase suggests that, in repair-proficient cells, these pathways can compensate for each other to some degree, and that inhibition of either pathway alone is insufficient to strongly enhance killing by TMZ.

We screened clinical-grade small molecules for HR/MMEJ inhibition and identified three hits. Two of these, Birabresib and Panobinostat, previously demonstrated favorable safety in solid tumors but showed little efficacy as single agents in GBM. Both agents cross the blood-brain-barrier and show additive killing with TMZ in GBM cell lines and xenografts[63,64]. However, Birabresib suppresses MMR, which is required for TMZ-induced fork collapse, and Panobinostat enhances MGMT activity, consistent with previous reports that prolonged treatment with HDACi promotes the evolution of TMZ resistance[25]. Therefore, these agents are likely not as promising in combination therapy with TMZ. Conversely, AZD1390 shows promise in combination with DNA-damaging therapy. It is currently being tested as a radiosensitizer in recurrent GBMs[6]. Our data suggest that additional combination therapies, such as AZD1390/TMZ, are possible, even in GBMs with intrinsic or acquired TMZ resistance caused by low-level expression of MGMT (G43 PDX), partial loss of MMR (si*MSH2*), enhanced HR/MMEJ (si*LIG4*) or multiple pathways at once (G22-TMZ, enhanced HR/MMEJ and partial loss of MMR). We note that TMZ/ AZD1390 combination was not appreciably toxic to MGMT-proficient T98G cells, suggesting that this combination would have a favorable therapeutic window when treating *MGMT*-hypermethylated GBMs.

Multiple investigations have implicated ATM in HR, but its role in MMEJ has been less clear, at least partially due to difficulties in multiplexing DSBR measurements. Cells from ataxia telangiectasia patients, which harbor mutations in *ATM*, display elevated MMEJ activity, suggesting that ATM suppresses MMEJ[65]. However, ATMi suppresses Pol theta foci formation, and mutation of ATM-dependent phosphorylation sites on CtIP suppresses MMEJ activity[45,49], suggesting that ATM promotes MMEJ. Using plasmid-based and genomically-integrated reporter assays, we show that ATM inhibition by AZD1390 or KU60019 suppresses both HR and MMEJ, especially in *TP53* mutant cells. Additionally, ATM knockout in U251 impairs HR and MMEJ (Fig. S14). Mechanistic studies revealed that ATMi suppress CPT- and radiation-induced RPA staining, as well as the phosphorylation of nibrin and Rad50 (Fig. 6). Taken together, these data suggest that ATM promotes MMEJ by activating DSB end resection, and that ATMi suppress this activity. Interestingly, AZD1390 inhibits the phosphorylation of end protection proteins, including Rif1, 53BP1, and Mettl16, suggesting that the removal of end protection barriers may be another mechanism by which ATM promotes HR/MMEJ. Whether ATM regulates MMEJ by additional mechanisms remains to be investigated, particularly in the context of ATM targets identified in this study. For instance, we detected ATM phosphorylation sites on MDC1 and TOPBP1 (Fig. 6I), both of which recruit Pol theta to DSBs[49]. Whether these phosphorylation sites—or others identified in our study—affect MMEJ activity remain to be determined. We also note that MMEJ inhibition by ATMi is affected by cell cycle phase, as AZD1390 does not affect MMEJ in G1-arrested cells but suppresses MMEJ in cells arrested in S or M phase. Additional work is needed to understand the molecular mechanism for ATM-independent MMEJ in G1, particularly because—although end resection is minimal in G1-arrested U251 cells— it is ATM-dependent (Fig. 6C, D).

Finally, we report molecular evidence underpinning the enhanced response of *TP53*-mutant GBMs to ATMi in combination with DNA-damaging agents. Previous investigations showed greater efficacy of ATMi/radiation or ATMi/doxorubicin in *TP53*-mutant cancers compared to *TP53*-WT[6,53,54], possibly because when p53 is absent, ATM is required for cell cycle arrest following DNA damage[53]. Our report expands upon this model by showing that, compared to *TP53*-WT GBMs, those with *TP53* mutation display elevated HR/MMEJ activity,

upregulated expression of ATM-dependent DSBR pathway genes, and robust activation of ATM-dependent HR/MMEJ following DNA damage. These signatures are consistent with an enhanced reliance upon ATM for DSBR in *TP53*-mutant GBMs. Taken together with evidence that p53 loss or mutation enhances expression of HR and MMEJ genes and causes replication stress and defective fork restart[56,66,67], we suggest that this contributes to enhanced ATMi response in *TP53*-mutant GBM.

Additionally, we show that the G2/M checkpoint is defective in *TP53*-mutant GBMs but intact in *TP53*-WT (Fig. 8). Both WT and mutant GBMs undergo transient G2 arrest after radiation/AZD1390 or TMZ/ AZD1390, presumably due to the presence of unrepaired DSBs in G2 phase. However, *TP53*-mutant GBMs—or a *TP53*-WT GBM expressing p53DD—fail to sustain G2 arrest and continue to enter mitosis, which they fail to complete, leading to cell death. Conversely, *TP53*-WT GBMs treated with radiation/AZD1390 or TMZ/AZD1390 show an accumulation of cells with 4N DNA content but a significant reduction in mitotic cells—consistent with G2 arrest—prior to eventual recovery and progression through mitosis into G1. These data suggest that an intact G2/ M checkpoint is a major protective mechanism against DNA damage in the presence of ATMi. The G1 checkpoint, although it is p53-dependent, does not appear to play as significant a role in protecting TP53-WT GBMs from DNA damage/ATMi combinations. *CDKN2A* and *CDKN2B* mutations are highly prevalent in GBM[34], suggesting that the G1 checkpoint may be perturbed even when *TP53* is unaltered. Additionally, ATMi suppress radiation-induced G1 arrest in the *TP53*-WT GBM cell line, A172 (Fig. 8B), similar to results in other *TP53*-WT cell lines[68], suggesting that even when the G1 checkpoint is intact, ATMi abrogate its initiation.

Taken together, we propose that dual inhibition of HR/MMEJ and the nibrin S343-dependent S phase checkpoint by ATMi leads to accumulation of treatment-induced DSBs in G2 phase. The integrity of the G2/M checkpoint is therefore a major determinant of susceptibility to DNA-damaging agents in combination with ATMi and represents an important targetable vulnerability in *TP53*-mutant GBM. Furthermore, we suggest that an intact G2/M checkpoint may safeguard *TP53*-WT tissues from combination therapy regimens employing ATMi in combination with DNA-damaging agents, potentially widening the therapeutic window.

In summary, we report a multiplexed DSBR assay which has the potential to inform oncology investigations and broaden the scope of research into DNA repair mechanisms. We highlight the utility of this approach by detecting clinically-relevant DSBR alterations in TMZ-resistant GBMs and identifying small-molecule HR/MMEJ inhibitors that potentiate TMZ in cell lines and PDX samples. Finally, we expand the understanding of the role of ATM in DSBR and present molecular evidence underlying the enhanced sensitivity of *TP53*-mutant cancers to ATMi in combination with DNA-damaging therapy. Our results call for further investigation into therapeutically targetable DSBR alterations in treatment-resistant cancers and for future combination therapies involving ATMi and DNA alkylating agents in *TP53*-mutant cancers, including GBM.

## Methods
### Ethics statement
This research complies with all relevant ethical regulations. Animal studies were approved by Mayo Clinic IACUC (approval #A5204 and A30206). Glioblastoma PDX lines were previously reported[69] and were derived from tumor specimens obtained following informed consent from adult GBM patients with the approval of the Mayo Clinic Ethics Review Board (IRB# 07-007623).

### Cell lines
U251, SF295, SNB-75, Hs568T, UACC257, HCT116, SKMEL28 cell lines were purchased from the DCDT tumor repository (https://dtp.cancer.

gov/repositories/dctdtumorrepository/default.htm) at the National Cancer Institute. U87-MG, T-98G, and A-172 which were purchased from ATCC and the TK6 knockouts were purchased from the TK6 Consortium (https://www.nihs.go.jp/dgm/tk6.html). All immortalized cell lines used in this study were confirmed free of mycoplasma by MycoAlert mycoplasma detection kit (Lonza cat. No.: LT07-703) at least once during the study. GBM lines including U251, A172, SF295, T98G, and U87 were tested three times during the initial experiments, at the beginning and end of revisions. Additionally, U251, U2OS, U87, T98G were confirmed mycoplasma negative by PCR testing using universal mycoplasma detection kit (ATCC Cat. No. 30-1012K). U251, U-87MG, T98-G, A-172, and U2OS cell lines were cultured in DMEM high glucose with pyruvate (ThermoFisher catalog number 11995065) with 10% fetal bovine serum (FBS, ThermoFisher 10437-028). SF295, SNB75, Hs578T, UACC257, SKMEL28, and HCT116 cells were cultured in RPMI (ThermoFisher 11875-093) with 10% FBS. U251 DR-GFP and EJ2-GFP reporter cell lines were kindly shared by Mary Helen Barcellos-Hoff[40] and were cultured in DMEM with 10% FBS.

## PDX samples

Previously reported PDX lines were derived from tumor specimens obtained following informed consent from adult GBM patients with the approval of the Mayo Clinic Ethics Review Board (IRB# 07-007623)[69]. Clinical information about the patients from which these samples were derived is publicly available at (https://www.cbioportal.org/study/clinicalData?id=gbm_mayo_pdx_sarkaria_2019). PDX explant cultures were generated as previously described[25]. Briefly, tumors were mechanically disaggregated and plated on laminin (Engelbreth-Holm-Swarm murine sarcoma basement membrane, Sigma Aldrich, cat. no. L2020)-coated flasks overnight. G14, G22, G39, G43, and G59 were cultured in DMEM (ThermoFisher catalog number 11995065) media supplemented with 10% FBS and 1% penicillin/streptomycin. G12 and G12 sublines which were cultured in serum-free media (StemPro Neural Stem Cell Serum-Free Medium, ThermoFisher cat. no. A105090) in flasks or plates coated with laminin (Engelbreth-Holm-Swarm murine sarcoma basement membrane, Sigma Aldrich, cat. no. L2020).

The patient sex and treatment status before sample collection were as follows:

G12 (Mayo-PDX-Sarkaria-12): Male primary tumor, untreated before PDX

G14: (Mayo-PDX-Sarkaria-14) Male recurrent tumor, radiotherapy, and Gefitinib prior to PDX

G22 (Mayo-PDX-Sarkaria-22) Male primary tumor, untreated before PDX

G39 (Mayo-PDX-Sarkaria-39) Male primary tumor, untreated before PDX

G43 (Mayo-PDX-Sarkaria-43) Male primary tumor, untreated before PDX

G59 (Mayo-PDX-Sarkaria-59) Female primary tumor untreated before PDX

Ages range from 51 to 83 years old.

## Chemicals

Temozolomide (Cat. No. S1237), AZD1390 (Cat. No. S8680), KU60019 (Cat. No. S1570), novobiocin sodium salt (Cat. No. S2492), Birabresib (Cat. No. S7360), Panobinostat (Cat. No. S1030), AZD7648 (Cat. No. S8843), AZD8055 (Cat. No. S1555), Trametinib (Cat. No. S2673), Sorafenib (Cat. No. S7397), Buparlisib (Cat. No. S2247), Ibrutinib (Cat. No. S2680), veliparib (Cat. No. S1004), palbociclib HCl (Cat. No S1116), nocodazole (Cat. No. S2775), and BO2 (Cat. No. S8434) were from SelleckChem. ART558 was from MedChem Express (Cat. No.: HY-141520). Aphidicolin was from MilliporeSigma (Cat. No. A0781). All stocks were prepared in DMSO (except for palbociclib HCl which was prepared in PBS) and stored in single-used aliquots at −80 °C.

## Enzymes

All enzymes were purchased from New England Biolabs and used in the provided buffer.

## Antibodies

| Target | Source | Manufacturer | catalog no. | Application | Conditions |
|---|---|---|---|---|---|
| Pol theta | Mouse | Millipore Sigma | SAB1402530 | Western blot | 1:1000 overnight (o/n) 4 °C in PBST 5% milk |
| ATM | Rabbit | Cell Signaling Technologies (CST) | #2873 | Western blot | 1:1000 o/n 4 °C in PBST 1% milk |
| CtIP | Rabbit | CST | #9201 | Western blot | 1:1000 o/n 4 °C in PBST 5% milk |
| Mre11 | Rabbit | CST | #4895 | Western blot | 1:1000 o/n 4 °C in PBST 2% milk |
| Lig4 | Rabbit | CST | #14649 | Western blot | 1:1000 o/n 4 °C in PBST 2% milk |
| BLM | Rabbit | CST | #2742 | Western blot | 1:1000 o/n 4 °C in PBST 2% milk |
| MSH2 | Rabbit | CST | #2017 | Western blot | 1:2000 2 h room temp (RT) in PBST |
| vinculin | Rabbit | CST | #4650 | Western blot | 1:1000 o/n 4 °C in PBST 5% milk |
| GAPDH | Mouse | Santa Cruz Biotechnologies | clone 0411 sc-47724 | Western blot | 1:2000 o/n 4 °C in PBST |
| cleaved PARP Asp214 | Rabbit | CST | #5625 | Immunostaining | 1:200 1 h RT in BD PermWash buffer |
| Replication protein A (RPA32/RPA2) | Rabbit | Abcam | ab76420 | Immunostaining | 1:200 1 h RT in BD PermWash buffer |
| Phospho-histone H3 Ser10 Alexafluor647 | Rabbit | CST | #3458 | Immunostaining | 1:50 1 h RT in BD PermWash buffer |
| goat anti-rabbit IgG Alexfluor 488 | Goat | Invitrogen | A11008 | Immunostaining | 1:200 1 h RT in BD PermWash buffer |
| goat anti-rabbit IgG Alexfluor 594 | Goat | Invitrogen | A11012 | Immunostaining | 1:200 1 h RT in BD PermWash buffer |

## Generation of plasmid-based host-cell reactivation assays

Promoterless Cherry (ΔCMV Cherry) was created by PCR amplifying pMax Cherry with a primer containing a 5′-tail with a NotI restriction site. Forward primer: 5′- GCC AGC GGC CGC TTA ATT AAG GCG GGC CAC GCG TCC TAG GAC CAG GTG GCC GGC CCG ATC GTC ATG ACG TAC GTC GAC TGA TCA TCA CAG GTA AGT ATC AAG GTT AC and reverse primer: 5′-GGA AGC GGC CGC CAT GCA TGG GAG GAG ACC GG were used in PCR with Phusion polymerase. The PCR product was gel purified and extracted using Monarch Gel Extraction Kit (New England Biolabs), digested with NotI, circularized with T4 DNA ligase, and transformed into DH5α E. coli (Invitrogen), which were plated on LB + kanamycin agar. Colonies were selected, plasmids amplified and isolated by Mini prep kit (New England Biolabs), and sequence confirmed by Sanger sequencing (Genewiz). After sequence confirmation, plasmid was amplified using Giga Prep Kit (Invitrogen Cat. No. K210009XP).

PspOMI Cherry was generated from pMax Cherry by site-directed mutagenesis using pMax Cherry (10 ng) as a template. Forward primer: 5′- CCC TCA GTT CAT GTA CGG GCC CAA GGC CTA CGT GAA GC and reverse primer: 5′-GCT TCA CGT AGG CCT TGG GCC CGT ACA TGA

ACT GAG GG were used at 500 nM final concentration in a 50 μL reaction using Phusion polymerase. DpnI (1 μL, 20 units) was added and incubated for 3 h at 37 °C. An aliquot (2 μL) was transformed into DH5α *E. coli* followed by Mini prep, Sanger sequencing, and plasmid amplification as above. Plasmid (200 μg) was linearized by treatment with PspOMI (300 units) in a 250 μL reaction at 37 °C for 2 h. Complete reaction was confirmed by agarose gel and enzyme was removed by phenol-chloroform extraction (Ultrapure, freshly opened bottle) followed by sodium acetate/ethanol precipitation.

GFP_MMEJ6 and BFP_MMEJ8 were generated by restriction cloning (NheI and HindIII) using the pMax backbone and a gBlock synthesized by IDT. Transformation, mini prep, and Sanger sequencing were conducted as above, and plasmids were amplified by Maxi prep kit (Qiagen) and then digested with ScaI-HF enzyme to introduce a DSB followed by cleanup as above.

### Transfection of DSB reporter plasmids

Adherent cell lines were transfected with reporter plasmids using Lipofectamine 3000 (ThermoFisher Cat. No. L300015). Cells were seeded into 12-well plates at 40,000-50,000 cells per well and adhered overnight. In inhibitor experiments, duplicate wells were treated with vehicle (DMSO, 0.1% final volume) or inhibitor at the appropriate dose for 1–2 h. For X-irradiation experiments, cells were treated (4 Gy or mock-irradiated) using a RadSource RS-2000 system and then transfected 1 h later. One well was transfected with WT plasmids (Undamaged plasmid cocktail) while another well was transfected with DSB reporter plasmids (Damaged cocktail) by mixing P3000 reagent (3 μL) with plasmid cocktail (1.5 μL) in Opti-MEM medium (50 μL). This was combined with Lipofectamine 3000 reagent (2.8 μL) in Opti-MEM (50 μL) and incubated for 5 min at room temperature before gently pipetting the transfection mixture (100 μL) into the appropriate well. After 20–24 h, cells were collected by trypsinization and analyzed by flow cytometry. Compensation and gating were established using single color controls as described previously[24,70]. Fluorescent reporter proteins were detected using the following Attune NxT parameters (channel−excitation wavelength, filter): BFP (VL1 - 405, 450/40); AmCyan (VL2 - 405, 525/50); GFP (BL1 - 488, 530/30); mCherry (561, 620/15). Experiments were repeated thrice on separate days.

Suspension cell lines (TK6) were transfected with reporter plasmids using Neon NxT transfection system (ThermoFisher). Cells were counted, collected by centrifugation, washed with PBS, and resuspended in the provided R buffer. A portion of the cells (13.5 μL) was mixed with plasmid cocktail (1.5 μL) and 10 μL of this mixture was transfected by electroporation according to the Neon protocol using default settings (1400 V, 20 ms pulse width, 1 pulse).

Four-color experiments utilized BFP, GFP, Cherry, and AmCyan. In some cases, three-color experiments were conducted with AmCyan omitted (such as when only HR and MMEJ were measured and not NHEJ). A single transfection utilized 1.5 μL volume in TE buffer and contained the following amounts of each reporter plasmid:

Undamaged plasmid cocktail: 100 ng BFP, 100 ng GFP, 100 ng Cherry, 100 ng AmCyan 1000 ng deltaCMV carrier

Damaged plasmid cocktail: 100 ng BFP_NHEJ, 250 ng GFP_MMEJ6, 100 ng mCherry_HR, 100 ng Am Cyan, 1000 ng deltaCMV carrier.

Repair efficiency was calculated as described for other FM-HCR reporters and as described below[70].

$$\%Reporter\ Expression = 100 \times \frac{X}{Y}$$

$$X\ (DSBR\ plasmid\ mixture) = \frac{BFP\ NHEJ\ Count \times Mean\ BFP\ fluorescence\ Intensity}{AmCyan\ Count \times Mean\ AmCyan\ Intensity}$$

$$Y\ (wild\ type\ cocktail) = \frac{wild\ type\ BFP\ Count \times Mean\ BFP\ fluorescence\ Intensity}{AmCyan\ Count \times Mean\ AmCyan\ Intensity}$$

GFP_MMEJ6:

$$X\ (DSBR\ cocktail) = \frac{GFP\ MMEJ\ Count \times Mean\ GFP\ fluorescence\ Intensity}{AmCyan\ Count \times Mean\ AmCyan\ Intensity}$$

$$Y\ (wild\ type\ cocktail) = \frac{wild\ type\ GFP\ Count \times Mean\ GFP\ fluorescence\ Intensity}{AmCyan\ Count \times Mean\ AmCyan\ Intensity}$$

Cherry_HR:

$$X\ (DSBR\ cocktail) = \frac{mCherry\ HR\ Count \times Mean\ mCherry\ fluorescence\ Intensity}{AmCyan\ Count \times Mean\ AmCyan\ Intensity}$$

$$Y\ (wild\ type\ cocktail)$$
$$= \frac{wild\ type\ mCherry\ Count \times Mean\ mCherry\ fluorescence\ Intensity}{AmCyan\ Count \times Mean\ AmCyan\ Intensity}$$

### FM-DSBR in arrested cells

For cell cycle arrest experiments, U251 cells were seeded at 100,000 cells per well and allowed to adhere overnight. The next day, six wells were treated with DMSO vehicle control (0.1%) or drug at the following concentration: Palbociclib (1.5 μM), nocodazole (300 nM), aphidicolin (1 μg/mL). After 18 h, three wells were treated with DMSO vehicle (0.1%) and three with AZD1390 (100 nM) for 1 h followed by transfection with FM-DSBR reporters as above. One well for each condition was left untransfected and used as a control in flow cytometry to ensure there were no false-positive fluorescent events. The next day, 20–24 h after transfection, cells were collected and analyzed by flow cytometry as described above.

### DR-GFP and EJ2-GFP MMEJ assays

U251 DR-GFP and EJ2-GFP reporter cell lines were reported previously[40]. We isolated stable GFP-negative subclones of the reporter cell lines by limiting dilution of a heterogenous population of transduced cells and subsequent expansion of cultures that were GFP-negative in the absence of SceI expression. For experiments, cells were seeded into 12-well plates at 40,000 cells/well. The following day, cells were treated with DMSO or drug (final concentration of DMSO 0.1% in all conditions) for 2 h prior to transfection with 500 ng pCBASceI plasmid[71] (Addgene plasmid #26477) and 10 ng pMax BFP plasmid as a transfection control. Lipofectamine 3000 (1.8 μL of Lipofectamine and 2 μL P3000 reagent, 100 μL Opti-MEM media) was used for transfection. After 72 h, cells were collected and analyzed by flow cytometry as above. Cells treated with DMSO or drug but lacking pCBASceI plasmid (replaced with carrier plasmid encoding a truncated, non-fluorescent protein) were included as negative controls. The number of GFP-positive cells was divided by the number of BFP-positive cells and multiplied by 100 to calculate normalized % GFP positive cells (with BFP-positivity used to account for differences in transfection efficiency between samples). Experiments were repeated thrice on separate days.

### Transient siRNA knockdown

U2OS or U251 cells were seeded into 6-well plates at 100,000 cells/well. The following day, cells were transfected with siRNA (1 μL, 10 pmol) from Dharmacon/Horizon using 3 μL Lipofectamine RNAiMax (ThermoFisher Cat. No. 13778075) in 100 μL Opti-MEM (ThermoFisher Cat. No. 11058021). After 72 h, cells were trypsinized and counted and then used for the appropriate assay. For FM-HCR assays, cells were seeded into 12-well plates at 40,000 cells/well, allowed to adhere overnight, and then transfected with reporter plasmids by Lipofectamine 3000 as described above. For clonogenic survival assays, cells were seeded at 750 cells per well in 6-well plates and then treated with the indicated dose of TMZ (typically 0, 5, 10 μM) followed by media replacement after 96 h and growth until colonies of greater than 50 cells were

visible (typically 12–14 days). All experiments were repeated thrice on separate days and siRNA knockdown was validated by Western blotting, and in the case of *POLQ*, by both Western blotting and qRT-PCR. Western blot for POLQ knockdown is in Fig. 1, blot for si*LIG4* and si*MRE11* is in Fig. 2, blot for si*MSH2* is in Fig. S15, si*ATM* is in S16, and blot for si*MRE11*, si*RBBP8*, and si*BLM* is in Fig. 6.

The following siRNA sequences were used: POLQ (5′-GCC AAU GGU CUG AUC AAU CUU), Rad51 (5′-AAG CUG AAG CUA UGU UCG CCA UU-3′), BLM (5′-GCU AGG AGU CUG CGU GCC GAU U-3′), MSH2 (5′-UAU AAG GCU UCU CCU GGC AAU UU-3′), MRE11 (5′-GAG CAU AAC UCC AUA AGU AUU-3′), RBBP8 (5′-GCU AAA ACA GGA ACG AAU CU U-3′), LIG4 (5′-CGA CCU UUU AGA CUC AAU UdTdT-3′), ATM (5′-GGU CUA UGA UAU GCU UAA AdTdT-3′) or non-targeting siRNA. The last two nucleotides at the 3′-end of each siRNA correspond to the overhang sequence.

## RNA isolation and qRT-pCR
Cells were collected 72 h after siRNA knockdown, and RNA was isolated using RNA Mini prep kit (New England Biolabs). cDNA was prepared using SuperScript VILO IV Master Mix (Invitrogen) starting from 250 ng of total RNA and qRT-PCR was conducted using SYBR Green Master Mix (Invitrogen) using an Applied Biosystems Real Time PCR system with the following primers: TCA GAA GGA TTC C actin reverse: 5′- GTC CAG GGC GAC GTA GCA CAG CTT CTC. POLQ forward: 5′- GAA ATG CCC TCT CAG TAC TGC TTG G POLQ reverse: 5′- CCA TCT GCT CTC CCA AAG ATT TAG C. Relative gene expression was calculated by ΔΔCt method.

## Cell cycle profiling and immunostaining
GBM cell lines were seeded at 125,000 cells per well in a 6-well plate. The following day, cells were pre-treated for 1 h with DMSO vehicle (0.1%), AZD1390 (10 nM), OTX015 (100 nM), or Panobinostat (10 nM) followed by X-irradiation with 4 Gy using a Radsource X-ray cabinet. After 24 hr, cells were trypsinized, quenched with complete media, and pelleted by centrifugation ($500 \times g$, 5 min). Media was removed and cells were washed once with PBS and then resuspended in 200 μL PBS followed by dropwise addition of ice-cold 80% ethanol with vortexing. Fixed cells were stored at -20 C overnight and then resuspended in PBS with 2% FBS, 10 μg/mL propidium iodide, 10 μg/mL RNAse A and analyzed by Attune NxT flow cytometer using the YL1 laser. G1, S, and G2/M populations were established by gating. Experiments were repeated thrice on separate days.

For staining mitotic cells with anti-phospho histone H3 S10, 150,000 cells were seeded into a T25 flask and irradiated with 4 Gy or mock (0 Gy) in the presence of 0.1% DMSO vehicle or AZD1390 (10 nM) and collected at 24 or 48 h. For TMZ treatment, 100,000 cells were seeded and cells were collected at 72 h. Cells were washed once with PBS, fixed and permeabilized with Cytofix/Cytoperm kit (BD Biosciences Cat. No. 554714) and stained with 1:50 dilution of Alexafluor647-conjugated anti-phosho histone H3 S10 antibody (Cell Signaling Technologies) for 1 h at room temperature, followed by washing with 1 mL of PermWash buffer (BD Biosciences), resuspension in PBS containing 2% FBS, DAPI (1 μg/mL), and RNAse A (10 μg/mL). Cells were analyzed using the RL1 channel (637 nm excitation, 670/14 emission filter) for Alexafluor647 and VL1 channel (405 nm excitation, 450/40 emission filter) for DAPI using an Attune NxT flow cytometer.

For staining with anti-cleaved PARP, cells were treated in a similar fashion, except they were incubated with 1:200 dilution of anti-cleaved PARP (Asp214) for 1 h, washed twice with PermWash buffer (BD), and then incubated for 1 h with Alexafluor 488-conjugated anti-rabbit secondary for U251 or A172 cells and then resuspended in 2% FBS with propidium iodide and RNAse A. For G14-GFP/G14-p53DD, secondary antibody was Alexafluor 594-conjugated anti-rabbit secondary (Invitrogen) for G14-GFP/G14-p53DD.

## Generation of knockouts and stable cell lines
To generate ATM knockout cells, U251-Cas9 cells were generated by transducing U251 cells with lentiviral particles packaged using pLX-311-Cas9 (Addgene plasmid #118018) followed by selection with blasticidin; U251-Cas9 cells were subsequently transduced with lentiviral particles packaged with ATM gRNA (BRDN0001149033, Addgene plasmid #77531) followed by clonal selection and expansion under puromycin selection.

To generate G10-GFP/G10-p53DD and G14 -GFP/G14-p53DD lines, the cDNA encoding the dominant negative p53 miniprotein, p53DD (amino acids 300-393), was excised from T7-p53DD-pcDNA plasmid, obtained from Addgene (Plasmid #25989). The cDNA for GFP was excised from pGIPZ and this was replaced with the cDNA for p53DD to generate the modified lentiviral vector, pGIPZ-p53DD-puro which was packaged into lentiviral particles by cotransfection of HEK 293T cells with lentiviral vector and helper plasmids (psPAX2 and pMD2.G encoding Gag/Pol and vesicular stomatitis virus glycoprotein, respectively). G10 or G14 PDX cells were transduced with lentivirus in media containing 5 μg/ml polybrene (MilliporeSigma) and selected in 5 μg/ml puromycin.

## Western blotting
Approximately 1 million cells were lysed in NETN buffer (100 mM NaCl, 20 mM Tris-Cl (pH 8.0), 0.5 mM EDTA, 0.5% (v/v) Nonidet P-40) with Complete Protease Inhibitor Cocktail (Millipore Sigma Cat. No. 11836153001) by 10 min incubation on ice followed by passage through 25 G syringe 12 times. Protein concentration was determined by BCA assay with BSA as a standard. Approximately 30 micrograms of total protein was separated by 7.5% SDS-acrylamide gel for large proteins (Pol theta, BLM, ATM) and 10% for all others. Proteins were transferred onto nitrocellulose membranes using Tris-glycine buffer containing 10% methanol in a cold room. For Pol theta and ATM, transfer was overnight at 25 V. For all other proteins, transfer was at 100 V for 1 h, except for BLM which was 100 V for 2 h. Membranes were blocked for 1 h using 5% blotting grade nonfat dry milk (Bio-Rad) in phosphate-buffered saline containing 0.1% Tween-20 (PBST) and incubated overnight with primary antibody at 1:1000 dilution in PBST containing 5% non-fat milk, except for BLM which was incubated in 2% milk. Membranes were washed three times with PBST and signal was detected by incubating with HRP-conjugated secondary antibody at 1:1000 dilution at room temperature 1 h followed by detection with Clarity Western ECL Substrate (Bio-Rad Cat. No. 1705061) and imaging with Invitrogen iBright 1500. SuperSignal Plus West Pico Reagent (ThermoFisher Cat. No. 34577) was used for detection and visualization of Pol theta and BLM.

## Replication protein A (RPA) staining
U251 cells were seeded at 250,000 cells per well into three 6-well plates. The following day, two wells in each plate were pre-incubated for 1 h with 0.1% DMSO vehicle, 10 nM AZD1390, or 100 nM AZD1390. Following pre-incubation, a plate was treated with either 0.1% DMSO vehicle, 2 μM camptothecin (CPT), or 10 Gy X-rays using a Radsource RS-2000. After 1 h, cells were washed with PBS, trypsinized, collected, and duplicate wells were pooled. RPA staining was conducted as described[44]. Briefly, non-chromatin bound RPA was extracted by resuspending cells in 100 μL cold 0.2% Triton-X in PBS and incubating on ice for 10 min followed by washing with 2 mL of 2% FBS in PBS. Cells were fixed and permeabilized using BD Cytofix/CytoPerm (BD Biosciences), incubated with primary (anti-RPA32/RPA2, ab76420 at 1:200 dilution) for 1 hr, washed with Perm/Wash buffer (BD Biosciences), and incubated with AlexaFluor488-conjugated secondary antibody (goat anti-rabbit, A11008 Invitrogen) for 1 h at 1:200. Cells were washed with Perm/Wash buffer and resuspended in PBS containing 2% FBS with propidium iodide (10 μg/mL) and RNAse A (20 μg/mL) and analyzed by flow cytometry using BL1 channel (488 nm excitation, 530/30 filter) for

excitation of AlexFluor488 and YL1 channel (561 nm excitation, 585/16 filter) for propidium iodide. Experiment was conducted thrice on separate days.

For ATM knockdown, U251 cells were seeded at 100,000 cells per well in a 6-well plate. Two wells were transfected with 15 pmol of non-targeting siRNA (siNT) or ATM-targeting siRNA (siATM) using Lipofectamine RNAiMax (4.5 µL per transfection). After 48 h, cells were trypsinized and counted and three wells of a 6-well plate were seeded with siNT cells or siATM cells (100,000 cells per well for each). After an additional 48 h, cells were treated with DMSO, CPT, or radiation and subjected to RPA staining as above. ATM knockdown was confirmed by Western blotting.

## Clonogenic survival assays

GBM cells were seeded into 6-well plates at 750 cells per well following gentle trituration to ensure usage of a single cell suspension. The following day, cells were pre-treated with the indicated drug for 1 h and irradiated with 1 Gy or 2 Gy X-rays using a Radsource RS-2000 Biological System. U251 cells were seeded into 12-well plates at 350 cells per well and pre-treated with the indicated concentration of each drug. Typical concentrations were AZD1390 (10 nM), OTX015 (50 nM or 100 nM), Panobinostat (2 nM), novobiocin (25 µM), BO2 (5 µM). Media was changed after 3 days to remove drug, and plates were kept in a humidified incubator for an additional 7 days followed by fixation with methanol for 20 min at room temperature and staining with a solution of 0.1% crystal violet in 25% methanol. Survival assays with TMZ were similar, except the inhibitors were co-administered with TMZ and treatment was for 4 days before media renewal. Experiments were repeated thrice on separate days. For survival assays following siRNA knockdown, cells were seeded into 6-well plates at 100,000 cells per mL, transfected the following day with siRNA (10 pmol) using Lipofectamine RNAiMax (3 µL), collected 72 h later, counted, and assessed for viability using a Vi-Cell cell counter (Beckman Coulter) and then used for clonogenic survival assays.

## Relative viability assays

GBM cell lines were seeded at 750 cells per well while PDX cells were seeded at 1500 cells per well in an opaque white 96-well plate in 100 µL of DMEM media (1% pen/strep included for PDX cells) per well. The following day, triplicate wells were treated with a freshly prepared 20× solution of TMZ with or without inhibitor in phosphate-buffered saline. Typically, the final concentration of TMZ was 0, 10, 25, or 50 µM and the final concentration of inhibitor were AZD1390 (10 nM or 25 nM), ART558 (1 µM or 2.5 µM), BO2 (10 µM), NVB (25 µM). Viability was assessed by CellTiter Glo 2.0 (Promega Cat. No. G9242) at the indicated time point (at least 120 h after addition of drug) according to the manufacturer protocol. Experiments were repeated thrice on separate days.

## Fluorescence-multiplexed double strand break repair (FM-DSBR) analysis in PDX samples

Glioblastoma PDX lines at low post-explant passage (P2 or P3) were seeded into 12-well plates at 25,000 cells per well in DMEM with 10% FBS and 1% pen/strep. The next day, cells were pre-treated with AZD1390 (100 nM) or vehicle (DMSO, 0.1% final volume) for 2 h. Cells were transfected with the following reporters using Lipofectamine 3000: BFP_NHEJ, GFP_MMEJ6, BFP_MMEJ8, or Cherry_HR. After 4 h, media was removed and replaced with fresh complete media containing DMSO or AZD1390 as appropriate. The following day, typically 20–24 h after transfection, cells were analyzed by flow cytometry. Experiments were repeated thrice on separate days.

## Mouse xenograft experiments

Xenograft therapy evaluations were conducted in orthotopic tumor model according to a protocol approved by the Mayo Institutional Animal Care and Use Committee. Orthotopic xenografts were established in female athymic nude mice (Hsd: Athymic Nude-Foxn1 nu) aged 6–7 weeks obtained from Envigo (Harlan), Indianapolis, IN. Mice with established xenografts were randomized into treatment groups with n = 8 per group for GBM22, n = 9 per group for GBM59, or n = 10 per group for GBM22TMZ and G59TMZ studies, respectively the day before initiating treatments. PDX GBM22 represents tumor from a male and GBM59 was derived from female patient, animal studies were performed in female animals only. Mice with established orthotopic tumors were randomized and treated with placebo/sham RT, TMZ (66 mg/kg daily for 5 days) or RT alone (2 Gy twice daily for 5 days [20 Gy total]) as previously described[72]. Mice were observed daily by a technician blinded to the identity of the treatment. The maximum tumor size allowed by IACUC is -2500 mm$^3$ for subcutaneous xenografts. This size was used only for the propagation of PDXs and was not exceeded at any point. The endpoint for animals with orthotopic xenografts was death or euthanasia due to moribund state as determined by weight loss exceeding 20%, inability to reach food/water, immobility, hunched posture, lethargy, seizures, circling and/or paralysis as per IACUC.

## Exome sequencing of PDX samples

Whole exome sequencing (WES) was performed in the Mayo Clinic Medical Genome Facility as described[73]. Briefly, paired-end libraries were prepared and sequenced with SureSelect Human All Exon V5+UTRs (or V4+UTRs) kit from Agilent Technologies (Santa Clara, CA) on the Illumina Hiseq 2500 platform (Illumina Inc., San Diego, CA, USA). The WES reads were aligned to the Human Reference Genome Build 37 using Novoalign (version 3.02.04) with the following options: -x 5 -i PE 425, 80 -r Random --hdrhd off -v 120 (http://www.novocraft.com/) followed by realignment and recalibration using GATK (version 3.3.0) as per recommended Best Practices[74]. Variant calling was performed with GATK's HaplotypeCaller using established pipeline and common variants eliminated based on the minor allele frequencies (>0.01) available in the 1000 Genomes Project or Exome Aggregation Consortium (EXAC). Mouse sequencing reads were removed using Xenome prior to mutation calling[75]. Data were deposited in the NCBI Sequence Read Archive, available with the following accession numbers:

G22_parental Accession Number: SAMN40262497 https://www.ncbi.nlm.nih.gov/biosample/40262497 G22-TMZ Accession Number: SAMN40262497 https://www.ncbi.nlm.nih.gov/biosample/40262498 G59 parental Accession Number: SAMN40262497 https://www.ncbi.nlm.nih.gov/biosample/40262501 G59-TMZ Accession Number: SAMN40262497 https://www.ncbi.nlm.nih.gov/biosample/40262502

## Phosphoproteomic analysis

Three 5-layer 875 cm$^2$ flasks were seeded with SF295 cells (7.5 million cells per flask) in 100 mL of RPMI with 10% FBS. Once cells were ~70% confluent, two flasks were treated with vehicle (DMSO, 0.01%) and one with AZD1390 (100 nM) for 1 h. One vehicle-treated flask and the AZD1390-treated flask were irradiated with 6 Gy using a RadSource RS-200 and then collected 45 min later by removing media, trypsinizing, and washing with PBS (which took -15 min), followed by snap freezing in liquid nitrogen.

To the cell pellets were added lysis buffer (2% SDS, 150 mM NaCl, 50 mM Tris pH 7.4) supplemented with Halt Protease and Phosphatase Inhibitor Single-Use Cocktail, EDTA Free. Homogenization was performed using an Omni International homogenizer, and insoluble material was removed by centrifugation at 3500 g for 15 min. The cleared lysates were reduced with 5 mM dithiothreitol (DTT) for 30 min at 37 °C. The reduced lysates were cooled to room temperature before alkylation with iodoacetamide (25 mM) for 30 min in the dark, whereupon the reaction was quenched by adding DTT (25 mM final). Proteins were purified by methanol/chloroform precipitation, the resulting protein pellet was incubated in freshly prepared 8 M urea in

digestion buffer (200 mM EPPS at pH 8.5) for 30 min at 37 °C. Digests were carried out in 1 M urea by further dilution with digestion buffer supplemented with 2% acetonitrile (v/v). LysC (2 mg/ml stock, enzyme-to-substrate mass ratio of 1:50) was added and samples were incubated at 37 °C for 4 h. Trypsin (enzyme-to-substrate mass ratio of 1:100) was added and the samples were incubated over night at 37 °C.

**TMT labeling.** For proteomic analysis, 60 μg of material was directly labeled with TMT 10plex reagents following the manufacturer's instructions. (Thermo Fisher Scientific). Labeling efficiency and TMT ratios were assessed by mass spectrometry, while labeling reactions were stored at −80 °C. When a TMT labeling efficiency of >95% was achieved, the reactions were quenched with hydroxylamine to a final concentration of 0.5% (v/v) for 10 min. The TMT labeled peptides were acidified with formic acid, pooled (as judged from ratio check data), and then solvent was evaporated. Purification was performed using acidic reversed phase C18 chromatography. Peptides were then fractionated by alkaline reversed phase chromatography into 24 fractions.

**Phospho-enrichment.** For the collection of phosphoproteomic data, the remaining digests were acidified to a final concentration of 1% formic acid and 0.1% trifluoroacetic acid (v/v) prior to purification using acidic reverse phase C18 chromatography. The phospho-peptides were enriched using Fe-NTA columns (A32992; Thermo Fisher Scientific). Phosphopeptides were also labeled using TMT 10plex reagents and processed in the same manner as the whole proteome samples. The pooled phosphopeptides multiplex was fractionated by high pH reversed phase chromatography (Thermo Fisher Scientific #84868) using a 12-step gradient of increasing acetonitrile. The dried fractions were further desalted by acidic C18 solid phase extraction (StageTip) and then finally re-suspended in 1% formic acid (v/v) for mass spectrometry analysis.

**Mass spectrometry analysis.** Data were collected using a MultiNotch MS3 TMT method using an Orbitrap Lumos mass spectrometer coupled to a Proxeon EASY-nLC 1200 liquid chromatography (LC) system (both Thermo Fisher Scientific). Samples were injected onto a 30 cm, 100 μm (internal diameter) column packed with 2.4 μm C18-AQ resin, with a needle tip of an internal diameter of ~5 μm provided by (ESI Solutions). Peptides were separated over a 2 h gradient from 7–28% acetonitrile in 0.125% formic acid, with a flow rate of 350 nl/min. First, an MS1 spectrum (Orbitrap analysis; resolution 120,00; mass range 400-1400Th) was taken, with dynamic exclusion time setting of 60 s. The MS2 spectrum was measured after collision-induced dissociation (CID, CE = 35) with a maximum ion injection time to maximum of 120 ms and multistage activation with a neutral loss of 97.9763 Da for phosphopeptides. For TMT quantification of peptides, SPS-MS3 was performed in the Orbitrap with a scan range of 100–1000 m/z, precursors were fragmented by high-energy collision-induced dissociation (HCD, CE = 55%). Injection time was set to a maximum of 300 ms Orbitrap resolution of 50,000 at 200 Th and MS isolation windows were varied depending on the charge state. Further details can be found in a previously published article[76].

**Data processing.** RAW files were converted into mzXML format and processed using a suite of software tools developed in-house for analysis of large-scale proteomics datasets. A Sequest-based in-house software was used to search peptides against a human database (downloaded from Uniprot March 2021) including know common contaminants such as human keratins. The in-house data analysis suite was described in detail previously and may be licensed from Harvard Medical School (for licensing enquiries, contact Steven Gygi (steven_gygi@hms.harvard.edu))[77]. Spectra were searched in both forward and reversed orientation in a target decoy database strategy. A false discovery rate of 1% was set for peptide-spectrum matches

following filtering by linear discriminant analysis. The final false discovery rate for collapsed proteins was 1%. Searches were performed using a mass tolerance of 20 ppm for precursors and a fragment ion tolerance of 0.9 Da. A maximum of two missed cleavages per peptide were allowed. Oxidized methionine (+15.9949 Da) and in the case of phopshopeptide samples, phospho-Ser/Thr/Tyr (+97.9763 Da) residues were dynamically searched, along with static modifications for alkylated cysteines (+57.0215 Da) and the TMT 10plex reagents (+229.1629 Da) on lysines and the N-termini of peptides. A modified version of the Ascore algorithm was used to define the confidence of the assignment of the sites that were phosphorylated (Huttlin et al., 2010). Relative protein quantification required a summed MS3 TMT signal/noise (s/n) >150 over all TMT channels per peptide along with an isolation specificity >60%. More details on the TMT intensity quantification and certain parameters can be found in another recent publication.

**Pride data upload.** Proteomics raw data and search results were deposited in the PRIDE archive and can be accessed under ProteomeXchange accession number: PXD047837. Data from the volcano plot in Fig. 6 are included in the source data file.

### Analysis of publicly available gene expression data
For pathway analysis, The PanCancer Genome Atlas (TCGA) GBM dataset was analyzed in cBioportal. Genes that were expressed at a significantly higher level in *TP53*-mutant GBM (q < 0.05 in cBioportal) were identified by comparison in cBioPortal and the list of 2239 significantly upregulated genes was downloaded from cBioPortal in October 2023 and input into the National Institute of Health DAVID Bioinformatics resource (https://david.ncifcrf.gov/tools.jsp) and subjected to pathway analysis using the Wikipathways set. Pathways that were significantly enriched (p < 0.05 with Benjamini Hochberg correction) in this gene set were ranked by fold enrichment, with the top 10 pathways displayed in Fig. 7.

For survival plots, the TCGA GBM dataset was analyzed in cBioPortal. Patients were stratified by median gene expression for the desired gene, and survival was plotted in cBioPortal with statistical analysis done in cBioPortal. Data were downloaded and plotted in GraphPad Prism.

### Statistics and reproducibility
Statistical analysis was conducted using GraphPad Prism, except for Fig. 3G, which was analyzed in cBioPortal, and Fig. 7E which was analyzed in DAVID Bioinformatics Resource. The statistical test used for each analysis is described in the figure legends. Statistical analysis was conducted from three or more biologically independent experiments (the exact number is listed for each panel in the figure legend). Error bars show the std. dev. of at least three independent experiments, except in Fig. 7D where error bars show the range. No statistical method was used to predetermine sample sizes, no data were excluded from the analysis. Animal experiments were randomized and in mouse survival assays, investigators were blinded to sample identity and treatment status during outcome assessment.

### Reporting summary
Further information on research design is available in the Nature Portfolio Reporting Summary linked to this article.

## Data availability
Source data for Figs. 1–8 are available in the source data file. Proteomics raw data and search results are available in the PRIDE archive and can be accessed under ProteomeXchange accession number: PXD047837. Exome sequencing data were deposited in the NCBI Sequence Read Archive, available with the following accession numbers: G22_parental Accession Number: SAMN40262497G22-

TMZ Accession Number: SAMN40262497 https://www.ncbi.nlm.nih.gov/biosample/40262498 G59 parental Accession Number: SAMN40262497 https://www.ncbi.nlm.nih.gov/biosample/40262501 G59-TMZ Accession Number: SAMN40262497 https://www.ncbi.nlm.nih.gov/biosample/40262502. Source data are provided with this paper.

## Code availability

No new code was used in this manuscript. The data analysis pipeline for the mass spectrometry experiments was described previously and can be licensed from Harvard Medical School[77]. For licensing enquiries, contact Steven Gygi (steven_gygi@hms.harvard.edu).

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

## Acknowledgements

We are grateful to Mary Helen Barcellos-Hoff for providing U251 DR-GFP and U251 EJ2-GFP cell lines.

## Author contributions

Conceptualization: Z.D.N., J.N.S., D.J.L., S.K.G.; methodology, investigation: D.J.L., S.K.G., G.A.B., A.S.H., B.L.C., N.M.C., J.C., S.T.; writing: D.J.L., Z.D.N., J.N.S., S.K.G., G.A.B.; supervision: Z.D.N., J.N.S.; funding acquisition: Z.D.N., J.N.S. All authors have read and agreed to the published version of the manuscript.

## Competing interests

J.N.S. reports receiving commercial research grants from AbbVie, ABL Bio, ADC Therapeutics, AstraZeneca, Bayer, Black Diamond, Boehringer Ingelheim, Bristol Myers Squibb, Glaxo Smith Kline, Inhibrx, Karyopharm, ModifiBio, Otomagnetics, Rain Therapeutics, Reglagene, SKBP, Sumitomo Dainippon Pharma Oncology, and Wayshine. Z.D.N. is a co-inventor on a related patent (US 9,938,587 B2) and reports past unrelated sponsored research agreements with Pfizer Inc., Ensoma, Agios, and Intellia Therapeutics.
