## [Peer Review File · Nature Communications]

ATM Inhibition Exploits G2/M Checkpoint Defects and ATM-Dependent Double Strand Break Repair in TP53-Mutant GlioblastomaREVIEWER COMMENTS

Reviewer #1 (Remarks to the Author): Expert in glioblastoma, ATM inhibitors, TMZ resistance, and DNA damage

Recommendation

The manuscript presents an interesting approach which has potential to impact the DNA repair field substantially, however, the assay and key claims are lacking validation and claims of translational relevance are overstated based on the data. The FM-DSBR assay is a conceptually elegant approach and one of the main strengths of the manuscript is the dissection of various MMEJ mechanisms probed by the MMEJ4, MMEJ6, and MMEJ8 reporters. Additionally, the identification of dual HR and MMEJ inhibitors as a therapeutic strategy has important implications for understanding DNA repair in treatment resistant cancers. However, there are several concerns with the validation of the authors claims and the penetrance of the observed molecular characteristics in other cell line models and GBM patients that warrant major revisions. The manuscript would be greatly strengthened by additional validation and discussion, outlined in the major and minor concerns below.

Summary

Laverty and colleagues report on a novel multiplex double strand break repair fluorescent reporter assay and its application to dissecting potential mechanisms of TMZ resistance in GBM. Most interestingly, the authors present reporters for both pol theta dependent MMEJ and pol theta independent MMEJ. Using these assays in combination with cell viability assays, they show that pol theta inhibition alone does not potentiate sensitivity to TMZ, which drives the researchers to look at inhibitors of multiple DSB repair pathways. The researchers identified three dual inhibitors that reduce both HR and MMEJ. AZD1390 (ATMi) shows increased radiosensitization and TMZ potentiation in the U251 model, likely due to its observed role in reducing end resection at double strand breaks. This effect was more pronounced and consistent in p53 mutant, TMZ resistant GBM models. In an effort to exemplify the utility of the FM-DSBR assay, they show that two TMZ resistant GBM models have defects in NHEJ and as a result are more sensitive to radiotherapy. The FM-DSBR assay is presented as a tool to study DSB repair related mechanisms of drug resistance. The manuscript ultimately identifies two potential therapeutic strategies for TMZ resistant GBM which require further study.

Major Concerns

1. Validation of MMEJ reporter assay controls

The authors claim that their various fluorescent reporter assays detect different MMEJ mechanisms, specifically pol theta dependent and pol theta independent MMEJ. They also utilize a siPOLQ cell line to support that the pol theta inhibitor ART558 is not effective at potentiating TMZ in their GBM models while the knock down is more effective. To fully support the use of the cell line models in Figure 1 and the MMEJ assays throughout the manuscript, more extensive validation of the pol theta expression in the generated POLQ knock down and knock out cell lines prior to the presentation of the MMEJ reporters is required in Figure 1. The validation of cell lines needs to include mRNA expression levels. Without proper validation of the pol theta expression, the validity of claims surrounding the MMEJ assays is not fully justified.

2. Validation of pol theta expression

The western blot showing pol theta knock down in Figure 4 needs to be validated by qPCR or other substantial antibody validation methods, as endogenous levels of pol theta protein are commonly too low for detection via Western blot in cell lines. This is evident in much of the pol theta literature, including Yousefzadeh et al. (2014), cited by the authors. This data should be included in the supplement.

3. Claim of HR/MMEJ driving TMZ resistance

In the results section discussing Figure 5, titled "AZD1390 potentiates TMZ and reverses acquired resistance driven by HR/MMEJ in TP53-mutant GBM PDX", the authors make the specific claim that the cases of TMZ resistance in their cell line models are driven by increased HR and MMEJ. They do observe increased HR and MMEJ as detected by their fluorescent reporter assays, however, there is

currently not enough evidence to claim that these effects drive the TMZ resistance in these cell lines. As this is the major finding derived from their novel assay with central importance to the overall impact of the manuscript, additional *in vitro* experiments with HR and MMEJ factor overexpression should be completed to isolate the effects of increased HR and MMEJ on TMZ sensitivity. The authors also present a LIG4^{-/-} and PRKDC^{-/-} TK6 model which could be probed for TMZ sensitivity in lieu of overexpression models, as these model shows increased HR and MMEJ in Figure 2. Increased resistance to TMZ in these two cell lines would support their claim. At present, the data suggests TMZ resistance may be associated with increased HR/MMEJ in MGMT null cases but lacks direct evidence.

Additionally, this claim is lacking discussion of other studies (or lack thereof) in the field of TMZ-resistance that support this claim. Commentary on the literature could provide additional justification and strengthen the authors' claim.

4. Translational prevalence of aberrant ATM activity

The authors suggest that the effect of AZD1390 is more consistent in p53 mutant cell lines, "possibly due to aberrantly high ATM activity". This statement should be verified by functional validation of ATM activity in the p53 WT and p53 mutant GBM cell lines. Data addressing this claim will give further mechanistic insight into sensitivity to AZD1390 and is important for consideration of molecular biomarkers. As the authors are proposing specific molecular contexts in which AZD1390 or other ATMis could be used, additional translational discussion should be included about observed ATM activity in GBM and TMZ-resistant GBM.

5. Lack of logical connection between major findings

Following with the points in major concern 3, the authors should also discuss the contribution of the dynamic balance between NHEJ and HR/MMEJ in the phenotypes they observe in cell lines sensitive to AZD1390 and RT. To the same end, the NHEJ reporter should be used to show relative NHEJ levels in the GBM cell lines in Figure 5C and 5D. This will give a more accurate indication of the penetrance of potential NHEJ defects across GBM models. This additional data will improve the concordance between the majority of the manuscript and the last figure without the suggested changes.

Minor Concerns

1. Figure 3C, 3D, and 3E should include reporter expression and cell survival data for ART558 treatment for purposes of comparison with NVB (another pol theta inhibitor) and the other dual inhibitors, which the authors claim are more efficacious in the Figure 3 results section.

2. ART558 should be evaluated via clonogenic survival assay as MMEJ is less frequently utilized in DSB repair proficient cells (see above). Furthermore, the clonogenic survival assays presented may be underrepresenting the efficacy of DSB repair disruption as they are 10-11 days in the manuscript. Increased incubation times allowing for >6 cell divisions should be utilized. Longer incubation times may also be necessary to observe differences in cell survival for pol theta inhibitors (NVB and ART558) in general.

3. The authors also observe defective NHEJ and show that these models are more sensitive to radiotherapy *in vivo*. Low vs. high NHEJ is a key part of the model presented in Fig. 6 as well. The authors should address whether this finding is consistent with what's observed in the clinic as TMZ + RT is standard of care.

Reviewer #2 (Remarks to the Author): Expert in DNA damage and repair, MMEJ, and homologous recombination

In this manuscript, the authors established fluorescence multiplexed double strand break repair (FM-DSBR) assay, a plasmid-based platform for simultaneous analysis of NHEJ, HR and MMEJ. They used this assay to determine the change of DSB repair pathways in glioblastoma (GBM) that

have acquired TMZ resistance and found that both HR and MMEJ are elevated. They also showed that ATMi could sensitize these TMZ-resistant (MGMT-defective) GBM to TMZ, and that acquisition of TMZ resistance is associated with defective NHEJ and radiosensitivity. This established FM-DSBR assay is useful for monitoring repair pathway change in tumor samples that can be transfected, but one caveat is that the reporters are not in chromatin. The findings are interesting and have clinical implications. However, mechanistic analyses need to be strengthened further. Elevation of HR and MMEJ in NHEJ-deficient cells in general was reported previously in the field.

1. The authors showed that ATM inhibition re-sensitizes TMZ-resistant GBM to TMZ, and reduces both HR and MMEJ through suppressing end resection. More detailed role of ATM inhibition in end resection needs to be addressed. Does ATM inhibit initial end resection by Mre11 and/or extensive end resection by BLM/DNA2 with Exo1? In addition to ATMi, ATMsh should also be used for end resection analysis. It is also not clear how ATM regulates and balances the use of HR and MMEJ pathways and what are the ATM downstream targets.

2. It is intriguing that ATM inhibits both POLQ-dependent MMEJ using GFP-MMEJ6 reporter (6 bp microhomologies 10 nt from the DSB) and POLQ-independent MMEJ using mOrange-MMEJ4 reporter (4bp microhomologies 3 nt from the DSB). Since 4 bp deletion with sticky end ligation is also common for NHEJ, DNAPKi and depletion or KO of NHEJ factors should be tested for MMEJ4. G22-TMZ and G59-TMZ are defective in NHEJ, and MMEJ4 should also be tested in paired TMZ cell lines.

3. Since PARPi does not inhibit MMEJ using plasmid MMEJ reporter, it needs to confirm POLQ-independent MMEJ4 in chromatin context.

4. AZD1390 inhibits HR/MMEJ in p53 defective but not so much in p53-proficient GBMs. This needs to be confirmed using isogenic pairs by depleting or KO p53 in p53-proficient GBMs. The possible mechanisms underlying this observation should be experimentally explored.

5. HDACi has a strong effect on both MMEJ and HR with a minor effect on NHEJ. Potentially, using HDACi is also a good treatment with TMZ for TMZ-resistant GBMs. Is the inhibition mechanism underlying HDACi different from ATMi?

General comments to reviewers:

We are grateful for the detailed and thoughtful feedback that we received from both reviewers. To address the concerns that were raised, we made extensive changes to the manuscript to further validate our FM-DSBR assay and to investigate treatment resistance in GBM. We also made significant progress in elucidating the molecular mechanism by which ATM inhibition suppresses repair and sensitizes cells to DNA damaging agents, as well as the basis for the differential response to ATMi in *TP53*-wild type and *TP53*-mutant GBM. Because of this, we have revised the title and added key points to the major conclusions of the paper. We believe that these changes greatly strengthen our manuscript and sincerely thank the reviewers for their insightful and constructive comments.

REVIEWER COMMENTS

Reviewer #1 (Remarks to the Author): Expert in glioblastoma, ATM inhibitors, TMZ resistance, and DNA damage

Recommendation

The manuscript presents an interesting approach which has potential to impact the DNA repair field substantially, however, the assay and key claims are lacking validation and claims of translational relevance are overstated based on the data. The FM-DSBR assay is a conceptually elegant approach and one of the main strengths of the manuscript is the dissection of various MMEJ mechanisms probed by the MMEJ4, MMEJ6, and MMEJ8 reporters. Additionally, the identification of dual HR and MMEJ inhibitors as a therapeutic strategy has important implications for understanding DNA repair in treatment resistant cancers. However, there are several concerns with the validation of the authors claims and the penetrance of the observed molecular characteristics in other cell line models and GBM patients that warrant major revisions. The manuscript would be greatly strengthened by additional validation and discussion, outlined in the major and minor concerns below.

Summary

Laverty and colleagues report on a novel multiplex double strand break repair fluorescent reporter assay and its application to dissecting potential mechanisms of TMZ resistance in GBM. Most interestingly, the authors present reporters for both pol theta dependent MMEJ and pol theta independent MMEJ. Using these assays in combination with cell viability assays, they show that pol theta inhibition alone does not potentiate sensitivity to TMZ, which drives the researchers to look at inhibitors of multiple DSB repair pathways. The researchers identified three dual inhibitors that reduce both HR and MMEJ. AZD1390 (ATMi) shows increased radiosensitization and TMZ potentiation in the U251 model, likely due to its observed role in reducing end resection at double strand breaks. This effect was more pronounced and consistent in p53 mutant, TMZ resistant GBM models. In an effort to exemplify the utility of the FM-DSBR assay, they show that two TMZ resistant GBM models have defects in NHEJ and as a result are more sensitive to radiotherapy. The FM-DSBR assay is presented as a tool to study DSB repair related mechanisms of drug resistance. The manuscript ultimately identifies two potential therapeutic strategies for TMZ resistant GBM which require further study.

Major Concerns

1. Validation of MMEJ reporter assay controls

The authors claim that their various fluorescent reporter assays detect different MMEJ mechanisms, specifically pol theta dependent and pol theta independent MMEJ. They also utilize a siPOLQ cell line to support that the pol theta inhibitor ART558 is not effective at potentiating TMZ in their GBM models while the knock down is more effective. To fully support the use of the cell line models in Figure 1 and the MMEJ assays throughout the manuscript, more extensive validation of the pol theta expression in the generated POLQ knock down and knock out cell lines prior to the presentation of the MMEJ reporters is required in Figure 1. The validation of cell lines needs to include mRNA expression levels. Without proper validation of the pol theta expression, the validity of claims surrounding the MMEJ assays is not fully justified.

We thank the reviewer for suggesting these experiments to enhance the rigor of our work. We added a Western blot showing POLQ knockdown in U251 (Fig. 1E) and validated knockdown with RT-qPCR as recommended (Fig. 2H). We validated TK6 POLQ knockout cell line by RT-PCR (Fig. S2) to confirm deletion of exons 20-22 as described in the source publication.

We provide additional validation of BFP_MMEJ8 using the pol theta inhibitor ART558 (Fig. 1B). We also validated GFP_MMEJ6 using pol theta inhibitors ART558 and NVB (Fig. 2D), knockdown of MRE11 and LIG4 (Fig. 2F), and knockdown of MRE11, RBBP8, and BLM (Fig. 6G).

2. Validation of pol theta expression

The western blot showing pol theta knock down in Figure 4 needs to be validated by qPCR or other substantial antibody validation methods, as endogenous levels of pol theta protein are commonly too low for detection via Western blot in cell lines. This is evident in much of the pol theta literature, including Yousefzadeh et al. (2014), cited by the authors. This data should be included in the supplement.

We agree that Pol theta expression levels are low in many cell lines and detection of Pol theta protein by western blotting has been difficult historically. We are confident we have successfully detected Pol theta using the mouse monoclonal antibody (SAB1402530), which was used to validate POLQ knockout in a previous report (PMID 34179826). We also initially had difficulty detecting Pol theta using standard chemiluminescent reagents, but we had much greater success using enhanced chemiluminescent reagents (SuperSignal picoECL), which we describe in the methods.

In addition to using this approach to validate POLQ knockout in U251 at the protein level by with western blot, we have confirmed the knockout at the cDNA level by RT-PCR (Fig. S2). We note that the shPOLQ SF295 cells no longer appear in this manuscript because, early in the revision process, we determined that mOrange_MMEJ4 was not in fact measuring theta-independent MMEJ and removed these data (more information about this appears below in response to comment 2 from reviewer 2).

3. Claim of HR/MMEJ driving TMZ resistance

In the results section discussing Figure 5, titled “AZD1390 potentiates TMZ and reverses acquired resistance driven by HR/MMEJ in TP53-mutant GBM PDX”, the authors make the specific claim that the cases of TMZ resistance in their cell line models are driven by increased HR and MMEJ. They do observe increased HR and MMEJ as detected by their fluorescent reporter assays, however, there is currently not enough evidence to claim that these effects drive the TMZ resistance in these cell lines. As this is the major finding derived from their novel assay with central importance to the overall impact of the manuscript, additional in vitro experiments with HR and MMEJ factor overexpression should be completed to isolate the effects of increased HR and MMEJ on TMZ sensitivity. The authors also present a LIG4^{-/-} and PRKDC^{-/-} TK6 model which could be probed for TMZ sensitivity in lieu of overexpression models, as these model shows increased HR and MMEJ in Figure 2. Increased resistance to TMZ in these two cell lines would support their claim. At present, the data suggests TMZ resistance may be associated with increased HR/MMEJ in MGMT null cases but lacks direct evidence.

We agree with these comments and undertook several approaches to identify whether enhanced HR/MMEJ directly contributes to TMZ resistance in GBM and to identify the clinical relevance of these observations.

- 1. To maximize physiological relevance, we depleted *LIG4* by siRNA in U251 and observed a protective effect against TMZ (Fig. 1K and 5D), as anticipated by the reviewer. We also found that, while TMZ/AZD1390 combination was effective in U251 siLIG4 cells, it was slightly less effective than in siNT cells, suggesting that enhanced HR/MMEJ partially undermines TMZ/AZD1390 combination (Fig. 5D). We compared siLIG4 and siMSH2 in a clonogenic survival assay (Fig. 1K) and found that depletion of MMR**

was a markedly stronger resistance mechanism, consistent with reports that loss of MMR is the major acquired TMZ resistance mechanism in the clinic (PMID: 21425258).

2. We measured FM-DSBR in an additional GBM PDX pair (G59 and G59-TMZ). Similarly to G22-TMZ, which was included in the original submission, G59-TMZ displayed enhanced HR/MMEJ. We had previously speculated that a mutation in *XRCC5* (encodes Ku80) may be responsible for treatment resistance in G59-TMZ. We knocked down Ku80 in GBM cell lines to interrogate whether this affected TMZ resistance, but this knockdown almost completely eliminated the ability of U251 cells to form colonies, preventing us from assessing the effect on TMZ sensitivity. Additionally, the G>A mutation in *XRCC5* in G59-TMZ results in a V405I mutation that is predicted to be tolerated (Table S1), suggesting that it is not an important contributor to TMZ resistance. We revisited the exome sequencing data from G59-TMZ and found damaging mutations in *RIF1* and *PPP1CC*, which encodes the catalytic subunit of the Rif1 effector, protein phosphatase 1 (Table S1). We have revised our manuscript (lines 192-195) to propose that these mutations impair the ability of these end-protection factors to suppress resection.
3. We analyzed the TCGA GBM dataset and found that expression of proteins in the 53BP1-Rif-shieldin pathway are associated with patient outcomes. In particular, high expression of the anti-resection protein SHLD1 and low expression of the pro-resection protein TRIP13, are associated with patient survival. We propose (lines 219-222) that these data are consistent with a model wherein increased utilization of resection-dependent DSBR pathways leads to poor GBM survival outcomes.
4. Finally, inhibiting DSB end resection by RBBP8 knockdown or ATMi markedly potentiated TMZ in GBM samples.

Taken together these data suggest that resection-dependent DSBR pathways contribute to TMZ resistance.

Additionally, this claim is lacking discussion of other studies (or lack thereof) in the field of TMZ-resistance that support this claim. Commentary on the literature could provide additional justification and strengthen the authors' claim.

We added additional commentary and references to the introduction (lines 53-63) to highlight that HR is a known TMZ resistance mechanism.

4. Translational prevalence of aberrant ATM activity

The authors suggest that the effect of AZD1390 is more consistent in p53 mutant cell lines, "possibly due to aberrantly high ATM activity". This statement should be verified by functional validation of ATM activity in the p53 WT and p53 mutant GBM cell lines. Data addressing this claim will give further mechanistic insight into sensitivity to AZD1390 and is important for consideration of molecular biomarkers. As the authors are proposing specific molecular contexts in which AZD1390 or other ATMis could be used, additional translational discussion should be included about observed ATM activity in GBM and TMZ-resistant GBM.

We agree that this statement was speculative and premature and have made substantial revisions, particularly in view of newly generated data. We had proposed that ATM activity may be "aberrant" when *TP53* is mutated because ATMi show stronger suppression of DSBR in *TP53*-mutant GBM than in *TP53*-WT. However, they also inhibit HR in some *TP53*-WT GBMs and in other cell lines such as HeLa and U2OS. Our new data provide a more detailed understanding of why p53 status is important for the cellular response to ATMi. Namely, we observe 1) upregulated expression of ATM-dependent DSBR genes in *TP53*-mutant GBM 2) robust activation of HR/MMEJ in *TP53*-mutant GBM (but not *TP53*-WT GBM) following genomic DNA damage, and 3) a defective G2/M checkpoint in *TP53*-mutant GBMs that causes them to enter mitosis with unrepaired DNA damage, leading to cell death. The

data supporting these points are detailed below. Although we believe these data are consistent with increased ATM activity in p53 mutant cancers, we do not have data to support the idea that increased ATM activity would have similar consequences in p53 wild type cancers. Indeed, the importance of a defective G2/M checkpoint in p53 mutant cancers suggests the opposite. We therefore do not believe ATM activity would be a reliable biomarker. Consequently, in our revised manuscript, we argue that p53 status holds the most translational potential as a biomarker for combination therapies including ATMi with DNA damaging agents.

1. We conducted pathway analysis of genes that are upregulated in TP53-mutant GBM and found that ATM-dependent DSB repair and ATM signaling were among the most strongly enriched pathways (Fig. 7E). Furthermore, we show that HR and MMEJ activity are higher in TP53-mutant GBM lines (Fig. 7D) and that these pathways are preferentially employed in response to DNA damage in TP53-mutant cell lines (Fig. 7F) and in isogenic cells where p53 transactivation activity is abolished (7G). We believe that these data support an increased reliance upon ATM-dependent HR and MMEJ in TP53-mutant GBM.
2. Irradiation with 4 Gy caused a robust increase in HR/MMEJ in TP53-mutant U251, SF295, and T98G cells but not TP53-WT U87 and A172 (Fig. 7F) and this was ATM-dependent (Fig S19A). Additionally, expression of dominant negative p53 protein, p53DD, in G14 cells caused these cells to more robustly enhance HR/MMEJ following irradiation (Fig. 7G).
3. The G2/M checkpoint is defective in TP53-mutant GBMs such that these cells cannot sustain G2 arrest after DNA damage (Fig. 8). Combination of AZD1390 with radiation or TMZ leads to a marked accumulation of mitotic cells, suggesting a failure to productively complete mitosis. TP53-wild type GBMs arrest in G2 phase and do not initially enter mitosis when treated with radiation and ATMi (Fig. 8D and 8F) but eventually progress through mitosis and into G1 48 hr after 4 Gy/ AZD1390, in contrast with TP53-mutant GBMs which remain in mitosis and stain positive for cleaved PARP, a marker of apoptosis.

5. Lack of logical connection between major findings

Following with the points in major concern 3, the authors should also discuss the contribution of the dynamic balance between NHEJ and HR/MMEJ in the phenotypes they observe in cell lines sensitive to AZD1390 and RT. To the same end, the NHEJ reporter should be used to show relative NHEJ levels in the GBM cell lines in Figure 5C and 5D. This will give a more accurate indication of the penetrance of potential NHEJ defects across GBM models. This additional data will improve the concordance between the majority of the manuscript and the last figure without the suggested changes.

We are particularly grateful for this comment, and we added data to Figure 7 showing NHEJ, HR, and MMEJ levels in p53-WT and p53-mutant GBMs (Fig. 7D), along with HR and MMEJ data in isogenic p53-altered pairs (Fig. 7H and Fig S18). We have also bolstered our discussion regarding the balance between NHEJ and HR/MMEJ (Lines 206-222, 522-526).

Minor Concerns

1. Figure 3C, 3D, and 3E should include reporter expression and cell survival data for ART558 treatment for purposes of comparison with NVB (another pol theta inhibitor) and the other dual inhibitors, which the authors claim are more efficacious in the Figure 3 results section.

We conducted additional experiments with ART558 as recommended. In Fig. 4D, we added ART558 to FM-DSBR in SF295 cells (this figure panel was previously Fig. 3E). We directly compared the effects of NVB and ART558 on DSBR in U251 (Fig. 2D). We also added clonogenic survival data for U251 with ART558/TMZ in Figure S3, where we show that ART558 does not potentiate killing by TMZ in U251, unlike Birabresib and AZD1390 (Fig. 5B).

2. ART558 should be evaluated via clonogenic survival assay as MMEJ is less frequently utilized in DSB repair proficient cells (see above). Furthermore, the clonogenic survival assays presented may be underrepresenting the efficacy of DSB repair disruption as they are 10-11 days in the manuscript. Increased incubation times allowing for >6 cell divisions should be utilized. Longer incubation times may also be necessary to observe differences in cell survival for pol theta inhibitors (NVB and ART558) in general.

We conducted clonogenic survival assays in U251 cells treated with ART558 and TMZ for 4 days followed by cell growth for an additional 10 days (Fig. S3). We did not observe potentiation of cell killing by TMZ under these conditions. We had previously explored NVB/TMZ combination at treatment times of up to 10 days, and we did not observe sensitization to TMZ under these conditions.

We agree that clonogenic survival assays are the gold-standard for assessing killing of cancer cells by DNA-damaging therapy. We note that the doubling time for this cell line is approximately 24 hours, allowing for well beyond 6 population doublings during the time of the assay. While a transient arrest is possible under treatment conditions, we reason that such an effect would enhance, rather than underrepresent the apparent cell killing since treated colonies would under these circumstances be more likely to remain below the threshold for counting in treated cells, but not the controls.

We would like to provide some context for the experimental conditions we chose. We had selected a treatment time of 4 days for clonogenic assays to give enough time for TMZ to form DSBs and then removed TMZ to give cells a chance to repair the DSBs. We assessed longer term TMZ combinations by viability assay instead of clonogenic survival because we found that treating cells with DNA repair inhibitors such as BO2 or ART558 for 7 days (followed by drug removal and additional 7-day outgrowth) substantially reduced colony forming efficiency (in the absence of TMZ). We were most interested in identifying drug combinations that showed synergistic effects, or where a non-toxic concentration of DNA repair inhibitor would markedly potentiate killing by TMZ or radiation (as is the case for AZD1390), so we did not rigorously analyze different doses and treatment times.

3. The authors also observe defective NHEJ and show that these models are more sensitive to radiotherapy in vivo. Low vs. high NHEJ is a key part of the model presented in Fig. 6 as well. The authors should address whether this finding is consistent with what's observed in the clinic as TMZ + RT is standard of care.

We added data and commentary about enhanced HR/MMEJ and its relationship to treatment resistance in the clinic (See Fig. 3G and lines 206-222). Based on our new data, we have also substantially revised our model and removed the graphic that emphasized NHEJ, previously Figure 6. We found that, in the TCGA GBM dataset, high TRIP13 expression is correlated with poor survival, suggesting that high levels of resection-dependent DSBR, and not defective NHEJ per se, are associated with poor prognosis. Further supporting this mechanism, low expression of the end protection factor SHLD1 is associated with poor survival. We did not exclude patients who receive radiation from this analysis, so most of the patients in this cohort who received TMZ also received radiation. This suggests that enhanced end resection may be compatible with resistance to both TMZ and radiation, compared with defective NHEJ—which may impart TMZ resistance (Fig. 1K and 3B) but also causes radiosensitivity (Fig. 3C)

Reviewer #2 (Remarks to the Author): Expert in DNA damage and repair, MMEJ, and homologous recombination

In this manuscript, the authors established fluorescence multiplexed double strand break repair (FM-DSBR) assay, a plasmid-based platform for simultaneous analysis of NHEJ, HR and MMEJ. They used this assay to determine the change of DSB repair pathways in glioblastoma (GBM) that have acquired TMZ resistance and found that both HR and MMEJ are elevated. They also showed that ATMi could sensitize these TMZ-resistant (MGMT-defective) GBM

to TMZ, and that acquisition of TMZ resistance is associated with defective NHEJ and radiosensitivity. This established FM-DSBR assay is useful for monitoring repair pathway change in tumor samples that can be transfected, but one caveat is that the reporters are not in chromatin. The findings are interesting and have clinical implications. However, mechanistic analyses need to be strengthened further. Elevation of HR and MMEJ in NHEJ-deficient cells in general was reported previously in the field.

1. The authors showed that ATM inhibition re-sensitizes TMZ-resistant GBM to TMZ and reduces both HR and MMEJ through suppressing end resection. More detailed role of ATM inhibition in end resection needs to be addressed. Does ATM inhibit initial end resection by Mre11 and/or extensive end resection by BLM/DNA2 with Exo1? In addition to ATMi, ATMsh should also be used for end resection analysis. It is also not clear how ATM regulates and balances the use of HR and MMEJ pathways and what are the ATM downstream targets.

We have done several experiments to provide further mechanistic insight into the role of ATM inhibition in end resection. The experiments are discussed briefly below, and in more detail in the main text on the indicated line numbers.

We assessed end resection in ATM-depleted cells, as recommended by the reviewer, and interestingly found that DNA damage-induced RPA signal was suppressed by ATM knockdown (Fig. S15), but to a lesser degree than ATM inhibition by AZD1390.

We conducted additional end resection experiments in U251 cells (lines 322-334) using camptothecin (CPT) because previous investigations showed that CPT induces a rapid and robust RPA signal that is essentially entirely dependent upon CtIP (PMID: 25629353). We compared this to irradiation with 10 Gy, which induced a lower degree of RPA signal. We found that ATMi suppressed CPT- and radiation-induced RPA signal (Fig. 6A and 6B). Based on the short time scale of this assay, these data suggest ATMi suppresses initial end resection in GBM cells. In support of the idea that this resection assay primarily reports short-range resection, Howard et. al. showed that CPT-induced RPA signal at 1 hr (the same timepoint we used) is highly dependent upon CtIP but less dependent upon DNA2 (PMID: 25629353). However, we cannot conclude from these data whether ATMi affect long-range resection.

We attempted to address how ATMi affect long-range versus short-range resection by knocking down end resection proteins Mre11, CtIP, and BLM, treating with AZD1390, and assessing HR/MMEJ (Fig. 6F-6H, lines 359-365). In these experiments, AZD1390 was additive with all knockdowns in suppressing HR efficiency, consistent with evidence that ATMi affect both initial resection and later stages of HR (PMC4381069). AZD1390 also appeared to have an additive effect on MMEJ, but knockdown of CtIP or Mre11 suppressed MMEJ to such low levels that it was difficult to measure the effects of inhibition precisely.

To further understand the mechanism of ATMi, we undertook an unbiased phosphoproteomic screen (lines 366-383) to determine which proteins are 1) phosphorylated within 1 hour after DNA damage (6 Gy X-rays) and 2) blocked by AZD1390. We detected known ATM targets including nibrin S343, nibrin S615 and Rad50 S635, consistent with a role for ATM in activating initial end resection. Interestingly, we detected many phosphorylation events on proteins that protect DSB ends, including 53BP1 and Rif1, as well as a recently described protein, METTL16 that protects DSB ends by sequestering Mre11 (PMID: 36138131).

We propose that our data, taken together, are consistent with a major role for ATM in the removal of end protection barriers to allow the initiation of end resection, which is required for both HR and MMEJ.

2. It is intriguing that ATM inhibits both POLQ-dependent MMEJ using GFP-MMEJ6 reporter (6 bp microhomologies 10 nt from the DSB) and POLQ-independent MMEJ using mOrange-MMEJ4 reporter (4bp microhomologies 3 nt from the DSB). Since 4 bp deletion with sticky end ligation is also common for NHEJ, DNAPKi and depletion or KO of

NHEJ factors should be tested for MMEJ4. G22-TMZ and G59-TMZ are defective in NHEJ, and MMEJ4 should also be tested in paired TMZ cell lines.

We are very grateful for this comment, as the proposed experiments helped us to determine that the mOrange_MMEJ4 reporter unfortunately does not measure theta-independent MMEJ. Instead this reporter is highly theta-dependent but also gives rise to theta-independent fluorescent events (which we believe to be HR events) under certain experimental conditions (detailed below). We have therefore removed the mOrange_MMEJ4 reporter from the manuscript.

We had planned to generate a genomic mOrange_MMEJ4 reporter. Before doing so, we characterized the plasmid-based reporter more fully. Depletion or knockout of LIG4 markedly increased mOrange_MMEJ4 expression, while depletion of Mre11, CtIP, or Exo1 suppressed mOrange_MMEJ4 expression. However, we determined that this reporter showed fluorescent events not only in the mOrange channel (YL1) but also in YL2 and YL3. The emission profile looked similar to mCherry, and critically, this fluorescence was present only when the carrier plasmid for the mCherry_HR assay was co-transfected (or a similar carrier plasmid with a truncation immediately downstream from the start codon). Both plasmids lack a promoter but contain the mCherry chromophore sequence, which we believe acts as a homology donor for the mOrange_MMEJ4 assay due to the sequence similarity between the two chromophores. Using stringent gating and compensation, we could distinguish the bona fide mOrange signal and found that this signal was markedly suppressed by ART558. The mCherry signal was unaffected by ART558 and disappeared entirely when other carrier plasmids were used. For these reasons, we believe that the reporter is not substantially different from existing MMEJ reporters in its dependence on POLQ to warrant the generation of a stably integrated system. Instead our evidence indicates that the MMEJ activity of this reporter is highly POLQ-dependent, but that confounding fluorescence from another repair pathway, likely HR, interferes with detection of MMEJ by mOrange_MMEJ4. Consequently, we did not pursue the remaining experiments proposed by the reviewer.

3. Since PARPi does not inhibit MMEJ using plasmid MMEJ reporter, it needs to confirm POLQ-independent MMEJ4 in chromatin context.

We agree with these comments about the importance of chromatin-based MMEJ assays, but as we determined that the mOrange_MMEJ4 assay does not specifically measure theta-independent MMEJ, we did not pursue this further. We note that U251-EJ2GFP cells treated with ART558 display substantial MMEJ activity (Fig. 4F), which may indicate the activity of theta-independent MMEJ in GBM.

We agree that additional work needs to be done to understand the requirements for theta-independent end joining, and we attempted several approaches for addressing this question. In view of our discovery regarding MMEJ4 and further challenges we encountered, we feel further analysis would be beyond the scope of the current manuscript. Briefly, we attempted to knock out POLQ in U251 to assess whether AZD1390 inhibits MMEJ that occurs in the absence of Pol theta. We have not been successful so far, but we plan to pursue this further in a future manuscript. Additionally, a recent report showed that polymerase lambda participates in a theta-independent MMEJ pathway (PMID: 36536104). We knocked down Pol lambda but did not observe any major effects on MMEJ efficiency using our plasmid-based assays (not shown), possibly because the effects are masked by the more dominant pol theta dependent repair. We plan to pursue this further in the future.

4. AZD1390 inhibits HR/MMEJ in p53 defective but not so much in p53-proficient GBMs. This needs to be confirmed using isogenic pairs by depleting or KO p53 in p53-proficient GBMs. The possible mechanisms underlying this observation should be experimentally explored.

We are grateful for this comment and have done several experiments including with isogenic cell lines to provide insights into the mechanism. We introduced data for additional TP53-WT GBM cell lines to Fig. 7B and 7C (G10 and G59) and compared DSB activity in TP53-WT and mutant GBMs (Fig. 7D, lines 413-422). We also note that

G14 data in Fig. 7C in the original submission were measured using BFP_MMEJ8 instead of GFP_MMEJ6, so we repeated experiments in G14 using GFP_MMEJ6 for the sake of consistency.

As suggested by the reviewer, we also employed isogenic pairs varying only in p53 status (Fig. 7G, 7H, and S18, lines 436-449) by using the previously published p53 dominant negative construct, p53DD (PMID: 8846919). This truncated protein inactivates wild type p53, presumably by forming non-functional oligomers with the wildtype protein. We found that p53DD expression enhanced HR and MMEJ activity in G10 and G14 PDX cells (Fig. 7H and Fig. S19). Additionally, G14 cells expressing p53DD showed a greater increase in HR/MMEJ in response to genomic DNA damage (Fig. 7G) and a greater ATM-dependency for these pathways (Fig. 7H).

We attempted additional isogenic comparisons using a previously published cell line (U87) expressing a dominant negative p53 protein (281G). Unfortunately, the cell lines we obtained were contaminated with mycoplasma, so we could not use these cells. We also attempted to knock out p53 in A172 cells, but we could not isolate any knockout clones, possibly due to poor colony forming efficiency in these cells.

We also show that ATM-dependent DSB repair pathways are upregulated in TP53-mutant GBMs in the TCGA GBM dataset (Figure 7E) and that TP53-mutant GBMs robustly activate HR/MMEJ in response to DNA damage, while TP53-WT GBMs do not (Fig 7F). To relate these findings to the mechanism of enhanced HR/MMEJ in TP53-mutant GBM, we added references and commentary to the Discussion (lines 565-574) indicating that loss or mutation in p53 leads to increased expression of HR and MMEJ genes (PMID 33385162, 16322760), increased replication stress (PMID: 29334356), and dysregulated origin firing (PMID: 28394262), which—in addition to a defective G2/M checkpoint (Fig. 8)—we propose drives the evolution of ATM-dependent HR/MMEJ to prevent cell death.

5. HDACi has a strong effect on both MMEJ and HR with a minor effect on NHEJ. Potentially, using HDACi is also a good treatment with TMZ for TMZ-resistant GBMs. Is the inhibition mechanism underlying HDACi different from ATMi?

We thank the reviewer for this question. We were also initially optimistic that this might be the case. However, we found that transient Panobinostat treatment enhances MGMT activity (Fig. 3C). We also added a reference showing that prolonged treatment with HDACi promotes evolution of TMZ resistance by reactivating MGMT expression, suggesting that HDACi would not be successful in combination with TMZ.

REVIEWERS' COMMENTS

Reviewer #1 (Remarks to the Author):

All concerns have been addressed in the revised manuscript.

Reviewer #2 (Remarks to the Author):

The authors have provided substantial new data and effectively addressed reviewers' comments. The manuscript is much improved. One minor point is on the suggestion that ATM may have roles other than promoting end resection in MMEJ (line 365). It will be interesting to further discuss this point in Discussion, particularly in connection with the ATM phosphorylation substrates identified in this revision.

We are grateful to the reviewers for additional comments on our revised manuscript. We have addressed the points that were raised below.

Reviewer #2 (Remarks to the Author):

The authors have provided substantial new data and effectively addressed reviewers' comments. The manuscript is much improved. One minor point is on the suggestion that ATM may have roles other than promoting end resection in MMEJ (line 365). It will be interesting to further discuss this point in Discussion, particularly in connection with the ATM phosphorylation substrates identified in this revision.

We agree with this point and have added the following to the Discussion to address reviewer 2's comment (change in red below).

Whether ATM regulates MMEJ by additional mechanisms remains to be investigated, *particularly in the context of ATM targets identified in this study. For instance, we detected ATM-dependent phosphorylation sites on MDC1 and TOPBP1 (Fig. 6I), both of which recruit Pol theta to DSBs [49]. Whether these phosphorylation sites—or others identified in our study—affect MMEJ activity remains to be determined.*